# Research on customer churn prediction and model interpretability analysis

**Ke Peng**[iD], **Yan Peng** *, **Wenguang Li**

College of Computer Science and Engineering, Sichuan University of Science and Engineering, Yibin, China

* 695078611@qq.com

## Abstract

In recent years, with the continuous improvement of the financial system and the rapid development of the banking industry, the competition of the banking industry itself has intensified. At the same time, with the rapid development of information technology and Internet technology, customers' choice of financial products is becoming more and more diversified, and customers' dependence and loyalty to banking institutions is becoming less and less, and the problem of customer churn in commercial banks is becoming more and more prominent. How to predict customer behavior and retain existing customers has become a major challenge for banks to solve. Therefore, this study takes a bank's business data on Kaggle platform as the research object, uses multiple sampling methods to compare the data for balancing, constructs a bank customer churn prediction model for churn identification by GA-XGBoost, and conducts interpretability analysis on the GA-XGBoost model to provide decision support and suggestions for the banking industry to prevent customer churn. The results show that: (1) The applied SMOTEENN is more effective than SMOTE and ADASYN in dealing with the imbalance of banking data. (2) The F1 and AUC values of the model improved and optimized by XGBoost using genetic algorithm can reach 90% and 99%, respectively, which are optimal compared to other six machine learning models. The GA-XGBoost classifier was identified as the best solution for the customer churn problem. (3) Using Shapley values, we explain how each feature affects the model results, and analyze the features that have a high impact on the model prediction, such as the total number of transactions in the past year, the amount of transactions in the past year, the number of products owned by customers, and the total sales balance. The contribution of this paper is mainly in two aspects: (1) this study can provide useful information from the black box model based on the accurate identification of churned customers, which can provide reference for commercial banks to improve their service quality and retain customers; (2) it can provide reference for customer churn early warning models of other related industries, which can help the banking industry to maintain customer stability, maintain market position and reduce corporate losses.

**Data Availability Statement:** All relevant data are available at: https://www.kaggle.com/datasets/sakshigoyal7/credit-card-customers.

**Funding:** This work was supported in part by the Key Laboratory of Enterprise Informationization

and Internet of Things Measurement and Control Technology in Sichuan Province Universities, No. 2021WYJ04; Sichuan University of Science and Engineering 2021 Postgraduate Innovation Fund Project, No. Y2021096. The funders had no role in study design, data collection and analysis, decision to publish, or preparation of the manuscript.

**Competing interests:** The authors have declared that no competing interests exist.

## 1 Introduction

With the rapid development of the Internet and the financial industry, Internet finance has become an emerging model in the financial industry and occupies an important position in market competition [1]. The development of the Internet continues to influence the development of the banking industry, on the one hand, financial products flow to the Internet, on the other hand, the profits of the traditional banking industry continue to decline, and the competition for the traditional fixed deposit business continues to increase. Customer retention and a customer-centric service approach have become the key to competitive advantage for any bank. In the traditional banking industry, the impact of Internet finance has diversified and personalized customer needs, resulting in the loss of a large number of existing customers. Customer churn is the behavior of a customer who, due to subjective or objective factors, terminates their current banking relationship for a period of time in favor of another bank. Customer churn is an important part of customer management system, it is understood that the cost of developing a new user is 5–6 times more than the cost of retaining an old customer [2], the user's choice determines the development of the enterprise, if we can predict the direction of customer churn through the factors affecting customer churn, and dig out the valuable content of potential lost customers, it will be necessary to help the banking industry to develop customer retention strategies and gain a competitive advantage. In recent years, for the customer churn prediction problem, scholars at home and abroad have used data mining techniques to analyze and establish customer churn prediction models, and applied classification algorithms to the field of customer churn, which is of great practical significance for enterprises to tap effective customers.

Many banks have used data mining tools to predict customer churn, and many of these cases have achieved good results. Burez [3] and Ying W [4] et al. argue that the problem of customer churn is a top priority for companies to deal with, and old customers can bring a greater profit return to the company, and the cost of attracting new customers is much higher than the cost of retaining old customers, and found that in the Shiv K. Sarin [5] argued that customer churn occurs at the transaction level, and found that when a company has the phenomenon of customer churn, it causes great losses to the company, on the one hand, they lose interest in buying the company's products and cause customer churn, and on the other hand, it also affects potential customers. Customers as the main body of market consumption, meeting their needs is the starting point of business management [1]. As a participant in the digital economy era, banks must start from the value of customers, seize market opportunities in advance, and continuously optimize products and technologies to improve the level of precision and wisdom in the marketing process, and provide customers with more reliable and convenient banking services. Therefore, banks need to anticipate customer churn, understand the causes of customer churn, and develop countermeasures in advance to extend the customer life cycle and reduce business losses [6].

This research aims to analyze and study the current status of customer churn in commercial banks and identify the causes of the current churn. The GA-XGBoost algorithm is used to build a bank customer churn prediction model and to explain and analyze the causes of customer churn as well as customer retention strategies by relying on interpretability related theories. Based on the study of data mining related technology theories and theories related to GA-XGBoost algorithm in customer churn application, the raw data of a bank is analyzed.

The main contributions of this paper are as follows:

1. First, by using multiple sampling techniques such as undersampling, oversampling and resampling to compare and contrast, the bank data imbalance problem is effectively solved.

2. Second, based on the traditional XGBoost algorithm for customer churn, an improved XGBoost model is proposed, the main idea of which is to improve the overall performance of the model through hyper-parameter optimization. Genetic algorithm is used to optimize the parameters of the composite XGBoost algorithm after data equalization to obtain the bank customer churn prediction model, and then the sample categories are divided. The experimental results show that the improved XGBoost model significantly improves the predictive ability.

3. Finally, the interpretable model is combined with the ensemble learning model, and the best prediction results of the ensemble learning model are interpreted globally and locally using the SHAP framework. This area is less analyzed by interpretable models, so this model can effectively help commercial bank decision makers to predict churning customers more accurately. In terms of its practical significance, when predicting bank customer churn, the SHAP value can visualize which characteristics mainly affect the customer churn outcome and whether the values of these characteristics increase the probability of customer churn, and then, based on the model prediction, facilitate bank staff to take relevant customer retention measures, which can further reduce bank customer churn, and also have certain reference significance for customer churn prediction in other fields.

The remainder of this paper is organized as follows. Section 2 describes related work and provides a literature review of existing applications of customer churn and interpretability analysis to big data. Section 3 analyzes the experiments, including (1) the GA-XGBoost model, (2) the SHAP interpretability framework, (3) data preprocessing and chi-square testing, and (4) the experimental design and evaluation metrics. Section 4 describes the analysis of the results. Section 5 provides a discussion of the research. Section 6 presents the conclusions and future work of this paper.

## 2 Literature review

### 2.1 Customer churn prediction

In recent years, for the problem of customer churn prediction, various scholars have used a combination of machine learning and data mining to analyze the real causes of customer churn and build appropriate churn prediction models to retain existing customers, among which support vector machine (SVM), decision tree, logistic regression and ensemble learning are widely used in this field for prediction and classification problems. Table 1 summarizes the literature review.

Literature [7] proposes the use of Kernel SVM to construct a churn prediction model for a telecom company, in addition, resampling techniques such as Tomek Link and ENN are used on the dataset to deal with unbalanced data. Literature [8] analyzes the sales of a retail store on Friday using a classification model and proposes a machine learning based technique to predict the age group of Friday shoppers for comparison. Machine learning algorithms such as decision trees, KNN, Naïve Bayes and neural networks were used to determine the most appropriate algorithm and the final decision tree achieved an accuracy of 88.22%. Literature [9] addresses the problem of class imbalance and focuses on comparing the performance of six oversampling solutions for handling CIP, demonstrating that MTDF and genetic algorithm based rule generation have the best overall prediction performance. In literature [10], a credit default prediction model was developed using GBDT and the K-means SMOTE oversampling method was used to address the imbalance in the data set, while the original hypothesis was rejected with a p-value < 0.001 using one-way analysis of variance, confirming the statistical significance of the improved performance of the proposed model. Literature [11] uses

**Table 1. The summary of the literature review.**

| Reference | Year | Model | Model interpretation | model optimization |
|---|---|---|---|---|
| Ly T V et al. [7] | 2022 | kernel SVM | O | Random Search |
| Shaukat K et al. [8] | 2021 | Decision Tree | O | O |
| Amin A et al. [9] | 2016 | Rough Sets | O | O |
| Alam T M et al. [10] | 2020 | GDBT | O | Random Search |
| Sana J K et al. [11] | 2022 | GDBT | O | Grid Search |
| Zhou J et al. [12] | 2019 | XGBoost | O | Grid Search |
| Kriti [13] | 2019 | XGBoost | Π | O |
| Maan J et al. [14] | 2023 | XGBoost | Π | Grid Search |
| Swetha P et al. [15] | 2020 | XGBoost | O | Grid Search |
| Lalwani P et al. [16] | 2022 | XGBoost | O | O |
| Our study | 2023 | GA-XGBoost | Π | Genetic Algorithm |

univariate techniques for feature selection in the customer churn domain and uses a grid search approach to select the optimal hyperparameters for the optimal model GDBT, demonstrating the benefits of applying data transformation methods and feature selection when training an optimized CCP model. Literature [12] proposes a default prediction model based on decision tree model using XGBoost model in integrated learning for accurate prediction of customer default in P2P lending, and also applies feature ranking based on learning model to P2P lending credit data with hyperparameter optimization for individual classifiers. Literature [13] uses the XGBoost algorithm in ensemble learning to predict customer churn in telecommunications and uses the LIME model to interpret the classifier prediction results with a maximum accuracy of 85%. The literature [14] identifies XGBoost as the churn prediction model. To improve the interpretability and transparency of the model, a new method is also proposed to calculate the Shapley values of the possible combinations of features to explain which features are the most relevant ones for the model to be highly interpretable, transparent and explainable to potential customers. In the literature [15], the Improvised-XGBoost churn prediction model was constructed for customer churn prediction in the telecommunication industry, and the loss function of the model was optimized, and the experimental results showed that the model performed better in terms of accuracy and precision, and had better results for some classes of sample prediction. In the literature [16], the K-fold cross-validation method was used to build the churn prediction model, and several common churn prediction algorithms were selected, including SVM, logistic regression, and decision tree, etc. Finally, it was concluded that the Adaboost model and the XGBoost model could achieve better results in terms of AUC with better classification performance.

Although the above studies have contributed to customer churn prediction, most of the current studies on customer churn prediction have used ensemble learning methods to construct customer churn models, for example, in the literature [10–16], ensemble learning has been used to construct the corresponding models. The ensemble learning approach, as a black box model with high complexity, cannot justify the prediction results of the models used. In the case of customer churn, the model can only determine the importance of the feature that influences customer churn, but cannot account for the positivity or negativity of the feature that influences customer churn on the prediction results. The SHAP algorithm is one of the latest interpretable methods that can explain both locally for a single sample and globally for the whole sample. Although the literature [13, 14] also used the same interpretable method to explain and analyze the constructed customer churn prediction model, they did not use innovative methods to optimize the model parameters, which resulted in no significant

improvement in the model's AUC and F1 results. In this study, instead of using grid search and random search for parameter optimization for the selected best model XGBoost, the GA-XGBoost model was constructed using genetic algorithm for parameter optimization. The experimental comparison results prove that the genetic algorithm is more effective compared to the traditional parameter optimization method, and the SHAP interpretable framework is used to transform the black box model of ensemble learning into an easy-to-understand visual mapping, which is convenient for bank staff to conduct in-depth research on customer churn.

## 2.2 Interpretability in data science

Some studies have already used SHAP interpretation methods in the fields of medicine, finance, and sports. For example, Yi Ming et al [17] proposed an interpretable model of public value in government new media that integrates XGBoost and SHAP model to explain and analyze the articles and comments under "Today's Headline", and obtained that the article topic type, public value type, article length, content form, etc. are important features that influence the consensus of articles in government headlines. The analysis shows that article type, public value type, article length, and content format are important features that influence the consensus of articles in government headlines. Chen, W. H. et al [18] interpreted the pre-lending overdue identification of Internet finance through the SHAP interpretation framework, visualized from the variable perspective and the sample perspective, respectively, to achieve the effect expressed by the XGBoost model. Yan Luo et al [19] demonstrated that the use of SHAP interpretable models helped clinicians to identify patients at risk of AKI in the ICU more accurately and quickly, and to provide better treatment to patients. AB Parsa et al [20] fused XGBoost and SHAP models to predict and characterize traffic accidents in real time. And X Wen et al [21] fused LightGBM and SHAP models to predict and characterize traffic accidents. Y Meng et al [22] used SHAP visualization to predict and characterize the usefulness of online reviews. Bin Liao et al [23] integrated the SHAP framework to analyze the important factors affecting player value at different positions based on the football player value prediction model to provide decision support for scenarios such as player value assessment, comparative value analysis, and player training strategy development.

In summary, since there are few studies in the field of bank customer churn using the SHAP method, this paper uses the SHAP method to analyze the interpretability of bank customer churn prediction results based on the bank customer dataset on the Kaggle platform, and explains the learning process of the model in terms of both local and global interpretation.

## 3 Material and methods

### 3.1 Technical route

This paper integrates SHAP interpretation framework and GA-XGBoost model to construct bank card customer churn prediction, which is mainly divided into the following steps: 1. Obtain public bank card customer churn dataset from Kaggle official website; 2.Perform data preprocessing, data cleaning, data conversion and other operations on the original dataset; 3. Deal with the problem of uneven distribution of datasets, including oversampling, undersampling, resampling and other methods for comparison; 4. Compare and verify the effect of six machine learning models and select the best model, perform genetic algorithm tuning on the best model, and then compare and verify the model effect; 5. Global and local interpretation of the best model prediction results using the SHAP framework. The technical route of this study is shown in Fig 1.

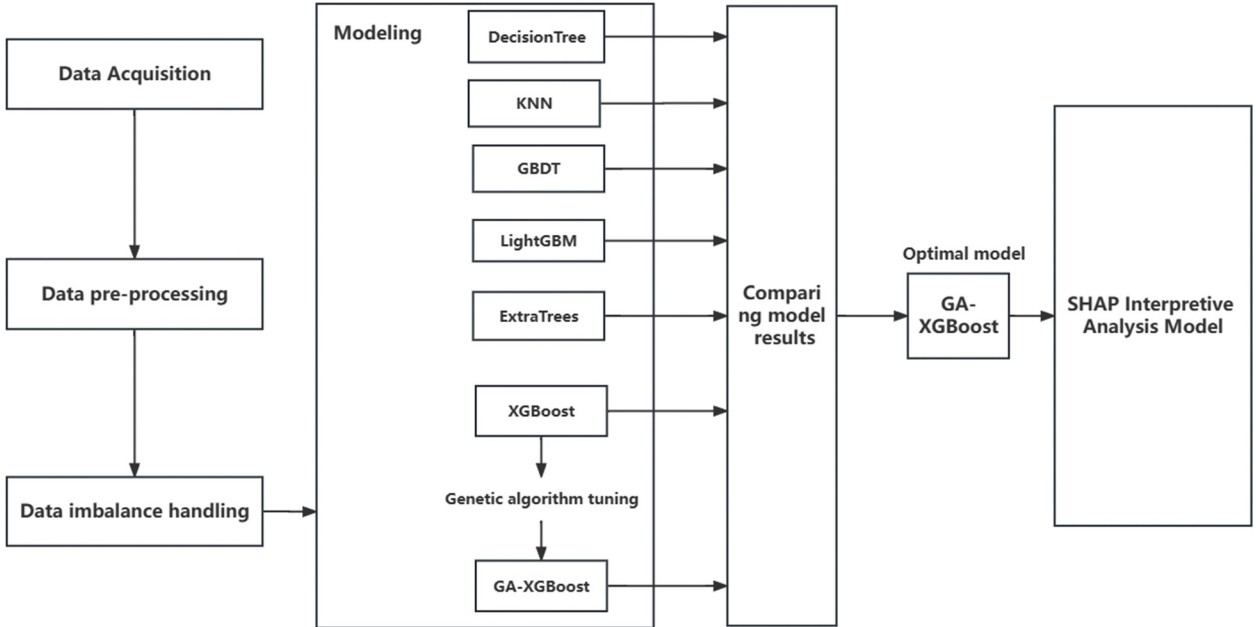

**Fig 1. Technical route.**

## 3.2 Related theories

**3.2.1 GA-XGBoost model construction.** In this paper, a bank customer churn model is constructed using a combination of genetic algorithm (GA) and XGBoost algorithm. XGBoost (Extreme Gradient Boosting) is a new method based on GBDT algorithm proposed by Chen in 2016 [24]. It is used as a boosting tree model to improve the traditional GBDT model, i.e., regularization and second-order Taylor expansion. The objective function of XGBoost for constructing bank customer churn model can be composed of loss function and regularization term. The specific customer churn prediction model objective function is shown in Eq (1), where $l(y_i, \hat{y}_i)$ represents the loss function of the predicted result and the true result; $\Omega$ represents the regularization term of the objective function, which is also represented as a penalty term to the complexity of the model [25].

$$L(\varphi) = \sum_{i=1}^{n} l(y_i, \hat{y}_i) + \sum_{k=1}^{K} \Omega(f_k) \tag{1}$$

The objective function obtained by second-order expansion using Taylor's formula is shown in Eq (2), where $g_i$ is the first-order derivative of $x_i$ and $h_i$ is the second-order derivative of $x_i$.

$$L(\varphi)^{(t)} \cong \sum_{i=1}^{n} [g_i f_t(x_i) + \frac{1}{2} h_i f_t^2(x_i)] + YT + \frac{1}{2}\lambda \sum_{j=1}^{T} \omega_j^2$$

$$\cong \sum_{j=1}^{T} \left[ \left(\sum_{i \in I_j} g_i\right) \omega_j + \frac{1}{2}\left(\sum_{i \in I_j} h_i + \lambda\right) \omega_j^2 \right] + YT \tag{2}$$

Derivative of ω and making the derivative of the derivative zero gives the ω that minimizes

the objective function, as shown in Eq (3).

$$\omega_j^* = -\frac{\sum_{i \in I_j} g_i}{\sum_{i \in I_j} h_i + \lambda} \tag{3}$$

$L(\varphi)_{min}$ is the minimum value of the objective function after the derivative of ω. The smaller the value of the objective function is, the better the tree model is, corresponding to the minimum value of the objective function as shown in Eq (4).

$$L(\varphi)_{min} = -\frac{1}{2}\sum_{j=1}^{T}\frac{\left(\sum_{i \in I_j} g_i\right)^2}{\sum_{i \in I_j} h_i + \lambda} + YT \tag{4}$$

The results of the split nodes of the tree model are counted as shown in Eq (5).

$$Gain = \frac{1}{2}\left[\frac{\left(\sum_{i \in I_L} g_i\right)^2}{\sum_{i \in I_L} h_i + \lambda} + \frac{\left(\sum_{i \in I_R} g_i\right)^2}{\sum_{i \in I_R} h_i + \lambda} - \frac{\left(\sum_{i \in I} g_i\right)^2}{\sum_{i \in I} h_i + \lambda}\right] - Y \tag{5}$$

Since the data set used in this paper is large and the XGBoost model has many parameters, the convergence is slow and the parameters have a great influence on the prediction results, and the traditional grid search method for parameter optimization has the problems of low accuracy, long computation time, and avoiding falling into local optimum when adjusting the parameters. Therefore, a new method of parameter tuning by genetic algorithm is used in this study. When performing hyper-parameter optimization, genetic algorithms can jump out of local extremes and use individual fitness functions to optimize the optimal parameter population, and continuously reduce the search area during the optimization process. Therefore, this paper proposes a GA-XGBoost model that combines genetic algorithm and XGBoost model, using the global search capability of genetic algorithm to achieve the optimal selection of XGBoost parameters, and using AUC as the fitness function to achieve the index adjustment. The flow of genetic algorithm is shown in Fig 2.

The GA-XGBoost optimal hyperparameter combination is the optimal number of chromosomes that the genetic algorithm can obtain when the number of iterations reaches the termination condition. In predicting bank customer churn, the three parameters of n_estimators, learning_rate and max_depth were optimized and initialized using the genetic algorithm. The genetic algorithm was used to obtain the next generation population and the best population was obtained by replicating, crossing and mutating in the parent population and then replacing the inferior population according to the fitness of the offspring population.

The steps for optimizing the XGBoost model based on genetic algorithm are specified as follows.

1. Initialize the XGBoost model parameters θ.

2. Continuous parameter tuning and training using genetic algorithm.

3. Use the AUC index as the fitness function, and select the optimal parameters according to the fitness.

4. Determine whether the maximum number of iterations is reached, if so, output the current training parameter value, otherwise return to (2).

5. Output the optimal parameter values and test the results in the test set.

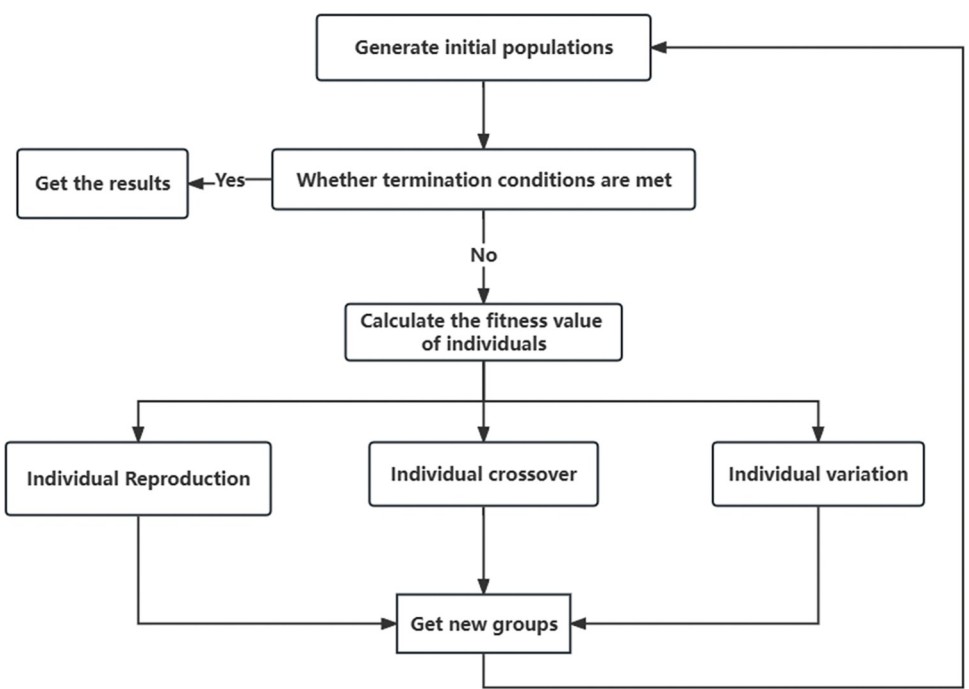

**Fig 2. Genetic algorithm flow chart.**

**3.2.2 SHAP interpretation framework.**   Based on Section 3.2.1, the GA-XGBoost algorithm can train a bank's customer churn prediction model with high accuracy, but due to the high complexity of the XGBoost model, as a black-box model, the interpretability is poor and the model cannot analyze how the sample eigenvalues affect the final prediction results. The bank wanted to understand the relationship between sample features and customer churn outcomes, and needed to analyze the correlation between feature importance and predicted churn outcomes. To solve the problem of poor interpretability of the GA-XGBoost model, the SHAP framework is introduced to analyze the reliability interpretation of the GA-XGBoost model prediction results.

SHAP (Shapley additive explanations) is an explanatory framework for explaining black box models, proposed by Lundberg in 2017 [26]. Using the optimal Shapley value in game theory, the model gives a prediction value for each prediction sample, and the SHAP algorithm can calculate the Shapley value of each feature as a measure of the contribution of each feature to the overall prediction result. SHAP interprets the prediction value of the model in two ways, the Shapley value of each input feature and the SHAP algorithm's analysis of each feature in terms of its importance in the overall sample [27].

The SHAP framework has powerful visualization capabilities, with the ability to show the interpretation of model predictions, and is widely used to interpret more complex classification and regression models. The traditional feature importance ranking method can only determine the importance of a feature and cannot explain how the feature affects the prediction results, while the main advantage of the SHAP value is that it can explain and analyze the degree of influence of each feature and also reflect the positive and negative influence of each feature. The results of the explanatory model of this framework are consistent with human intuition [28].

For GA-XGBoost model for customer churn classification, the model output is a probability value. Suppose $x_i$ is the i-th customer sample, where the j-th feature of $x_i$ is $x_{ij}$, $y_i$ is the model's

prediction of whether the sample is churned or not, $y_{base}$ is the mean value of all sample target variables, and $f(x_{ij})$ contribution of the j-th feature in the i-th sample to the final prediction $y_i$, the SHAP value is calculated as shown in Eq (6).

$$y_i = y_{base} + f(x_{i1}) + f(x_{i2}) + \cdots + f(x_{ik})$$ (6)

For the SHAP value of each feature, when $f(x_{ij}) > 0$, it means that the feature has a boosting effect on the prediction result, and when $f(x_{ij}) < 0$, it means that the feature makes the contribution lower. the biggest advantage of the SHAP framework is that it can correctly reflect the impact of each feature and the positivity and negativity of the impact.

### 3.3 Experimental data and its preprocessing

To test the effectiveness of the proposed method, the Credit Card Customer dataset, which consists of personal information, behavioural information and transaction information of bank customers, is taken from the Kaggle website. The specific dataset URL is https://www.kaggle.com/datasets/sakshigoyal7/credit-card-customers. The details of the characteristic variables in the dataset are shown in Table 2.

Fig 3 shows the variable correlation heat map, where the feature correlation heat map is plotted to explore and analyze the features that influence bank customer churn. The lighter color of the heat map indicates the lower correlation of two features, and the darker color indicates the higher correlation of two features. From the feature correlation heat map, we can analyze that the correlation between Avg_Utilization_Ratio and the average value of credit limit, total turnover balance, and available credit limit in the past year are -0.48, 0.63, and -0.54, respectively, and the correlation coefficient between them is large, so we decide to delete the "Avg_Utilization_Ratio" feature.

In this paper, One-Hot unique hot coding is introduced to process the category-type variables, and Marital_Status, Education_Level, and Card_Category are assigned to the coding.

**Table 2. Details of feature variables of the data set.**

| Feature Name | Feature values |
|---|---|
| CLIENTNUM | [708082083,828343083] |
| Customer_Age | [26,73] |
| Gender | Male, Female |
| Dependent_Count | Yes, No |
| Education_Level | Doctorate, Post-Graduate, Graduate, College, High School, Uneducated |
| Marital_Status | Divorced, Married, Single |
| Income-Category | Less than 40K, 40K-60K, 60K-80K, 80K-120K, 120K+ |
| Card_Category | Blue, Gold, Silver, Platinum |
| Total_Relationship_Count | [1,6] |
| Months_Inactive_12_mon | [0,6] |
| Contacts_Count_12_mon | [0,6] |
| Credit_Limit | [1438.4,34516] |
| Total_Revolving_Bal | [0,2517] |
| Avg_Open_To_Buy | [3,34516] |
| Total_Amt_Chng_Q4_Q1 | [0,3.397] |
| Total_Trans_Amt | [510,18484] |
| Total_Trans_Ct | [10,139] |
| Total_Ct_Chng_Q4_Q1 | [0,3.714] |
| Avg_Utilization_Ratio | [0,0.999] |
| Attrition | Attrited Customer, Existing Customer |

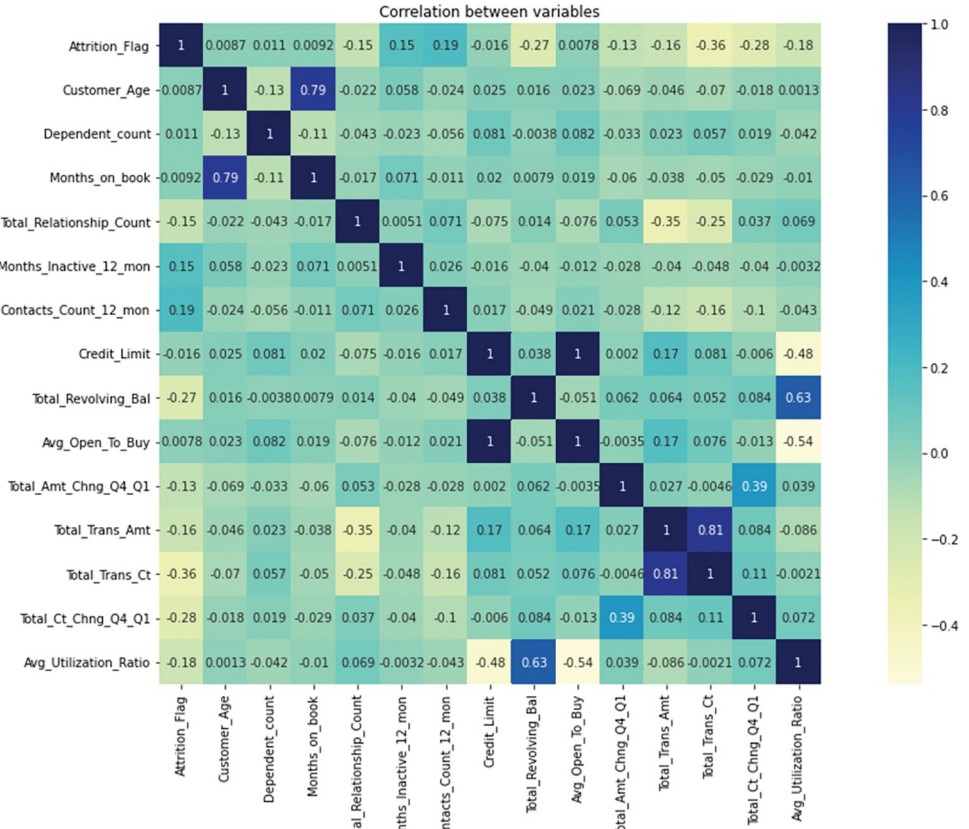

**Fig 3. Heat map of variable correlation.**

After the unique hot coding process, all the features of the data set are continuous type variables, and then this paper uses the Z-score normalization method to normalize the original data so that the feature data conform to the Gaussian normal distribution.

The chi-square test, which is a free distribution test, does not exist for specific parameters and the assumption of overall normal distribution, and the mean and variance cannot be calculated. For the bank customer dataset we used, we set $H_0$: the feature has no effect on customer churn results; $H_1$: the feature has an effect on customer churn results. Through the experimental results in Table 3, we can conclude that the p-values of Gender, Income_Category, and Education_Level are almost less than 0.05, and the original hypothesis $H_0$ is rejected, that is These features have a significant effect on customer churn results. p-value indicates the probability of mutual independence when the sample row variables and column variables are drawn from the overall, and since the probability value of p is small, we reject the original hypothesis that the features and customer churn results are independent of each other.

**Table 3. Chi-square test for selected features.**

| features | p-value of Chi Squared test |
| --- | --- |
| Gender | 0.00019635846717310269 |
| Dependent_count | 0.09150463456682643 |
| Marital_Status | 0.10891263394840227 |
| Income_Category | 0.025002425704390617 |
| Education_Level | 0.05148913147336627 |

**Table 4. Confusion matrix.**

| Actual value | Predictive value | |
|---|---|---|
| | **Churn** | **Not Churn** |
| Churn | TP | FN |
| Not Churn | FP | TN |

## 3.4 Experimental environment and evaluation index

In the experiments, the algorithm is programmed in Python 3.6 language using XGBoost, Imbalance, and Scikit-learn toolkit.

In the area of bank customer churn, due to the imbalance of dichotomous data in the dataset, the learning metrics for evaluating bank customer churn were chosen to focus the evaluation metrics on positive cases, where accuracy, recall, precision, and F1 score metrics were obtained from the confusion matrix, as shown in Table 4. In Table 4, TP indicates actual customer churn and predicted customer churn; FN indicates actual customer churn but predicted customer non-churn; FP indicates actual customer non-churn but predicted customer churn; TN indicates actual customer non-churn and predicted customer non-churn.

In the bank customer churn prediction problem, the evaluation metrics used based on the confusion matrix are accuracy, precision, recall, and F1-score, respectively. the specific formulas are shown in the following order.

$$\text{Accuracy} = \frac{TP + TN}{TP + TN + FN + FP} \tag{7}$$

$$\text{Recall} = \frac{TP}{TP + FN} \tag{8}$$

$$\text{Precision} = \frac{TP}{TP + FP} \tag{9}$$

$$\text{F1} - \text{score} = \frac{2*precision*recall}{precision + recall} \tag{10}$$

The AUC value is the area under the ROC curve, which is a curve plotted with TPR as the vertical axis and FPR as the horizontal axis. Where TPR = TP/(TP+TN) and FPR = FP/(FP+TN), the AUC value is suitable for the overall evaluation of the bank customer churn prediction model, and the closer the value is to 1, the better the performance of the bank customer churn prediction model.

## 4 Results

### 4.1 Unbalanced data set processing

The distribution of whether the data set in Fig 4 is churned or not can be analyzed to show that the percentage of churned customers is low, and the ratio of online customers to churned customers is about 5:1. The distribution of the data set is extremely unbalanced, customer data is easily predicted as majority class samples, which leads to poor prediction results. Using the XGBoost model for prediction, although the accuracy rate can reach 90%, the recall rate is only 38%, and the value of AUC is 0.69, indicating that the model classification effect is weak, and it easily predicts customer data as majority class samples, leading to poor prediction results. To solve the data imbalance problem, this paper uses multiple resampling algorithms

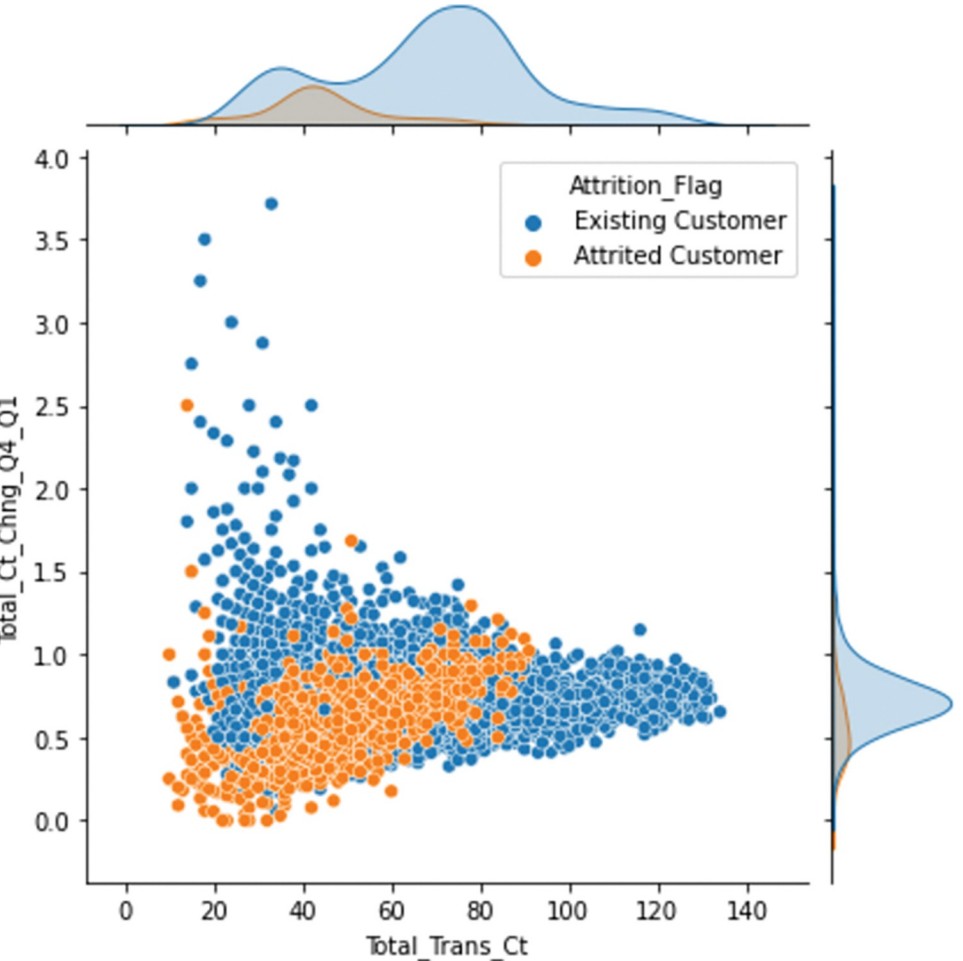

**Fig 4. Distribution of bank customer churn label.**

to achieve balanced processing by changing the distribution of data. In the experiments, the ADASYN and SMOTE algorithms of the oversampling method and the SMOTEENN algorithm of the hybrid sampling method are used, and the data processed by different sampling algorithms are substituted into the XGBoost model for training, and the results are evaluated by multiple evaluation metrics, and finally the algorithm with the best effect is selected to deal with the data imbalance. A comparison of the metrics of different sampling algorithms in the XGBoost model is shown in Table 5.

The SMOTE algorithm increases the number of minority class samples by manually synthesizing new samples, and its main method is to generate new samples by linear interpolation among the minority class samples. The ADASYN algorithm, on the other hand, synthesizes a

**Table 5. Performance comparison of different adoption algorithms in XGBoost model.**

| Sampling algorithm | Accuracy | Precision | Recall | AUC |
|---|---|---|---|---|
| No processing | 0.89 | 0.90 | 0.38 | 0.69 |
| SMOTE | 0.96 | 0.88 | 0.81 | 0.93 |
| ADASYN | 0.96 | 0.87 | 0.83 | 0.94 |
| SMOTEENN | **0.96** | 0.85 | **0.92** | **0.99** |

new minority class sample by interpolating a certain proportion of the minority class samples. Compared to SMOTE, the ADASYN method places more emphasis on the problem of discrimination difficulty of minority class samples, and more minority class samples can be generated around the positive class sample data. The SMOTEENN algorithm is a composite sampling technique combining the SMOTE and ENN algorithms, which first increases the minority class samples by linear interpolation using the SMOTE method, and then the majority class samples are removed by the ENN algorithm. The noisy data in the samples that are different from their majority K-nearest neighbor sample classes are removed by the ENN algorithm [29].

The experimental comparison shows that the evaluation metrics of several sampling algorithms outperform the unprocessed data samples, so the use of sampling algorithms to balance the data set is suitable for modeling bank customer churn. Comparing the three sampling algorithms, the SMOTEENN algorithm of the hybrid sampling method performs best with an AUC of 0.99, an accuracy of 0.96, and a recall of 0.92, which is better than the ADASYN and SMOTE algorithms of the oversampling method. Therefore, the best algorithm, SMOTEENN, is used to balance the bank customer data set.

## 4.2 Comparison of experimental results

In this paper, the SMOTEENN algorithm is used in combination with a machine learning classification algorithm. The XGBoost model is selected to compare the performance with five classification models, namely LightGBM, DecisionTree, KNN, GDBT and ExtraTrees, and the comparison results are shown in Table 6 and Fig 5.

Comparing the results in Table 6 and Fig 5, it can be seen that XGBoost model for predicting customer churn has the highest accuracy, precision, F1 value, and AUC of 0.9608, 0.8488, 0.8830, and 0.9902, respectively, among the six models, and only has a slightly lower recall than LightGBM's 0.9242. The XGBoost algorithm stores the input data as a block structure, and the block structure greatly reduces the computation and improves the prediction speed when predicting the input data. In addition, XGBoost can also perform second-order Taylor expansion on the objective function and add regularization terms, which can prevent the overfitting phenomenon. Compared with other comparative algorithms, XGBoost algorithm is more stable and the prediction results are more accurate than other algorithms.

## 4.3 Model hyperparameter optimization

The XGBoost model has various parameter types, and three parameters, n_estimators, learning_rate, and max_depth, which have a large impact on the model, are selected for tuning in this paper. The parameters are optimized by genetic algorithm, and AUC is used as the fitness function of genetic algorithm. The obtained tuning results are shown in Table 7.

According to the characteristics of the genetic algorithm and the range of XGBoost parameters, the tuning parameters of the genetic algorithm are set as follows: the population size is 50,

**Table 6. Comparison results of different model.**

| Classifier | Precision | Recall | F1-score | Accuracy | AUC |
|---|---|---|---|---|---|
| XGBoost | **0.8488** | 0.9201 | **0.8830** | **0.9608** | **0.9902** |
| LightGBM | 0.7058 | 0.9242 | 0.8004 | 0.9260 | 0.9824 |
| DecisionTree | 0.6008 | 0.9098 | 0.7237 | 0.8885 | 0.9504 |
| KNN | 0.4993 | 0.7705 | 0.6060 | 0.8391 | 0.8507 |
| GBDT | 0.8167 | 0.9037 | 0.8580 | 0.9520 | 0.9836 |
| ExtraTrees | 0.4756 | 0.7971 | 0.5957 | 0.8263 | 0.8934 |

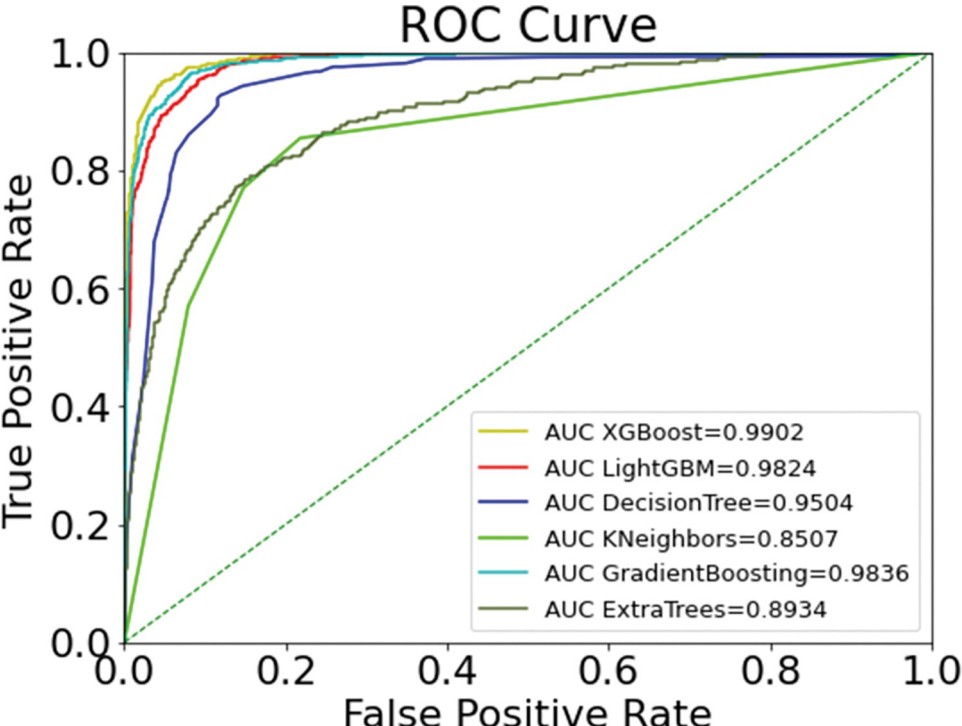

**Fig 5. Comparison results of each classification model.**

the maximum number of evolutionary generations is 10, the log information is recorded every second time, the judgment threshold of single-objective optimization into stagnation is 1e-6, and the maximum upper limit of the evolutionary stagnation counter is 10. As shown in Figs 6 and 7, after 10 iterations, the population optimal single objective function value converges to the population average single objective function value and searches for the optimal fitness function AUC of 0.9912. The final approximate optimal solution of GA-XGBoost is n_estimators = 265, learning_rate = 0.097, max_depth = 5. 5. The results of the XGBoost model after genetic algorithm tuning are shown in Table 8.

According to the comparison of model tuning indexes shown in Table 8, GA-XGBoost is the model after genetic algorithm tuning. The experimental results show that the GA-XGBoost model has improved in each evaluation index, and the precision, recall, F1 value, accuracy, and AUC are optimal, which are 0.8760, 0.9262, 0.9004, 0.9671, and 0.9912, respectively. The XGBoost model after parameter optimization using genetic algorithm performs optimally in terms of generalization performance and time consumption due to the proper combination of parameters.

### 4.4 Model performance analysis

The GA-XGBoost model optimized with XGBoost hyperparameters by genetic algorithm was used in the test set to predict bank customer churn, and the confusion matrix of the test set

**Table 7. Results of genetic algorithm tuning parameters.**

| Parameter | Parameter Category | Parameter Meaning | Default Value | Tuning results |
|---|---|---|---|---|
| n_estimators | Learning task | Number of sub-models | 100 | 265 |
| learning_rate | Booster | Learning rate of the model | 0.3 | 0.097 |
| max_depth | Booster | Maximum depth of the tree | 3 | 5 |

```
===============================================================================
gen|  eval  |    f_opt    |    f_max    |    f_avg    |    f_min    |    f_std
-------------------------------------------------------------------------------
 0 |   50   | 9.90998E-01 | 9.90998E-01 | 9.87804E-01 | 9.39260E-01 | 7.68758E-03
 1 |  100   | 9.90998E-01 | 9.90998E-01 | 9.90626E-01 | 9.90190E-01 | 2.22198E-04
 2 |  150   | 9.91501E-01 | 9.91501E-01 | 9.90902E-01 | 9.90718E-01 | 1.66010E-04
 3 |  200   | 9.91501E-01 | 9.91501E-01 | 9.91063E-01 | 9.90916E-01 | 1.65135E-04
 4 |  250   | 9.91519E-01 | 9.91519E-01 | 9.91202E-01 | 9.90998E-01 | 1.47092E-04
 5 |  300   | 9.91519E-01 | 9.91519E-01 | 9.91339E-01 | 9.91182E-01 | 1.18442E-04
 6 |  350   | 9.91519E-01 | 9.91519E-01 | 9.91452E-01 | 9.91330E-01 | 5.61394E-05
 7 |  400   | 9.91522E-01 | 9.91522E-01 | 9.91499E-01 | 9.91480E-01 | 6.20402E-06
 8 |  450   | 9.91522E-01 | 9.91522E-01 | 9.91504E-01 | 9.91501E-01 | 6.56842E-06
 9 |  500   | 9.91522E-01 | 9.91522E-01 | 9.91507E-01 | 9.91501E-01 | 8.74193E-06
```

**Fig 6. Iteration diagram of genetic algorithm.**

prediction results is shown in Fig 8. In the test set, the actual number of churned customers was 488, and the GA-XGBoost model judged that 452 customers were churned and 36 were misjudged, with a recall rate of 0.9262; the actual number of unchurned customers was 2551, and the GA-XGBoost model judged that 2487 customers were unchurned and 64 were misjudged, with an accuracy rate of 0.8760. The area under the ROC curve of GA-XGBoost is 0.9912, as shown in Fig 9. The GA-XGBoost model metrics are good by performance analysis.

## 4.5 Model comparison analysis

Finally, after the bank customer dataset was balanced by SMOTEENN, we built seven different machine learning models of LightGBM, DecisionTree, KNN, GBDT, ExtraTrees, XGBoost, and GA-XGBoost, respectively. The model comparison results are shown in Fig 10 and Table 9, where the precision, recall, F1 score, accuracy, and AUC of GA-XGBoost are 0.8760,

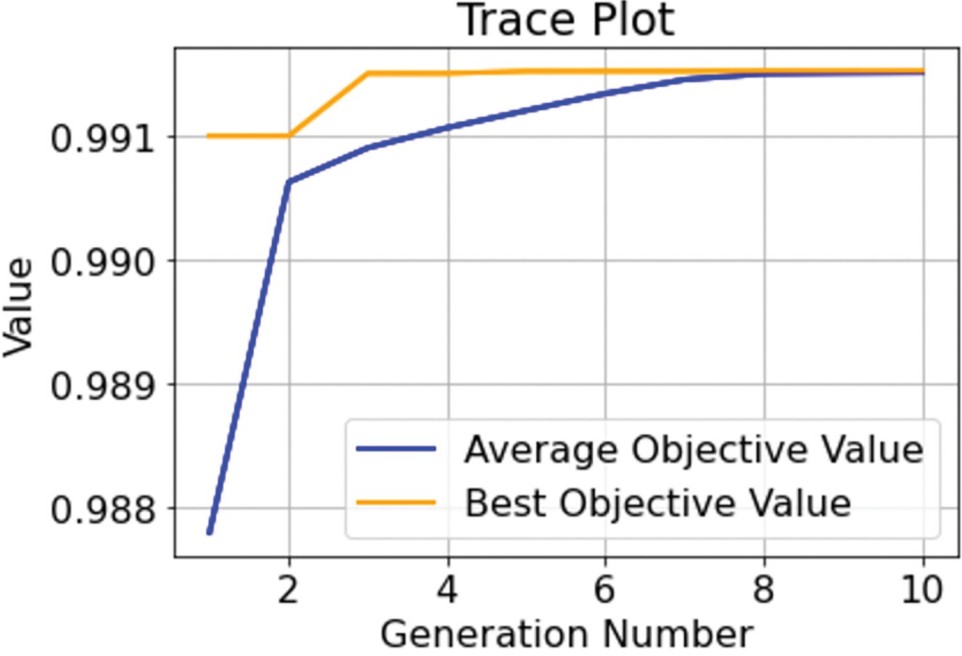

**Fig 7. GA-XGBoost optimization process diagram.**

**Table 8. Comparison of GA-XGBoost with XGBoost and LightGBM test results.**

| Classifier | Precision | Recall | F1-score | Accuracy | AUC |
|---|---|---|---|---|---|
| XGBoost | 0.8488 | 0.9201 | 0.8830 | 0.9608 | 0.9902 |
| LightGBM | 0.7058 | 0.9242 | 0.8004 | 0.9260 | 0.9824 |
| GA-XGBoost | **0.8760** | **0.9262** | **0.9004** | **0.9671** | **0.9912** |

0.9262, 0.9004, 0.9671, and 0.9912, respectively, and each evaluation index is the highest among the seven different models. Through the experimental comparison, GA-XGBoost is the most effective in predicting bank customer churn, which can effectively help commercial banks to deal with customer churn in advance and make corresponding retention plans in real life.

### 4.6 SHAP-based model interpretation

In this paper, we use the SHAP method to explain the learning process of the model, which is the degree of contribution of each feature to improve the prediction ability of the model. It is a more direct representation of the degree of influence of the features on the model. The focus of this chapter is on the explanatory analysis of the model prediction results of bank customer churn based on the SHAP framework.

**4.6.1 Global explanation.** The results of the GA-XGBoost model analysis using the SHAP method are shown in the summary plot of the feature analysis in Fig 11, where each row represents a feature and the horizontal axis is the SHAP value of the model, i.e., the impact of the

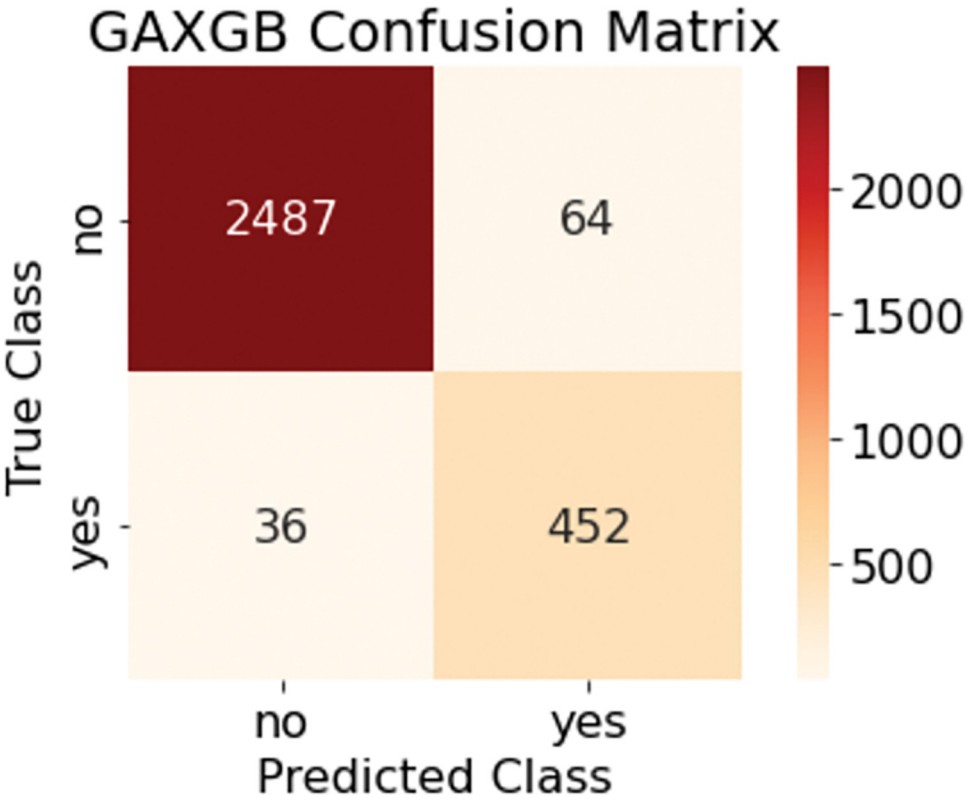

**Fig 8. GA-XGBoost confusion matrix.**

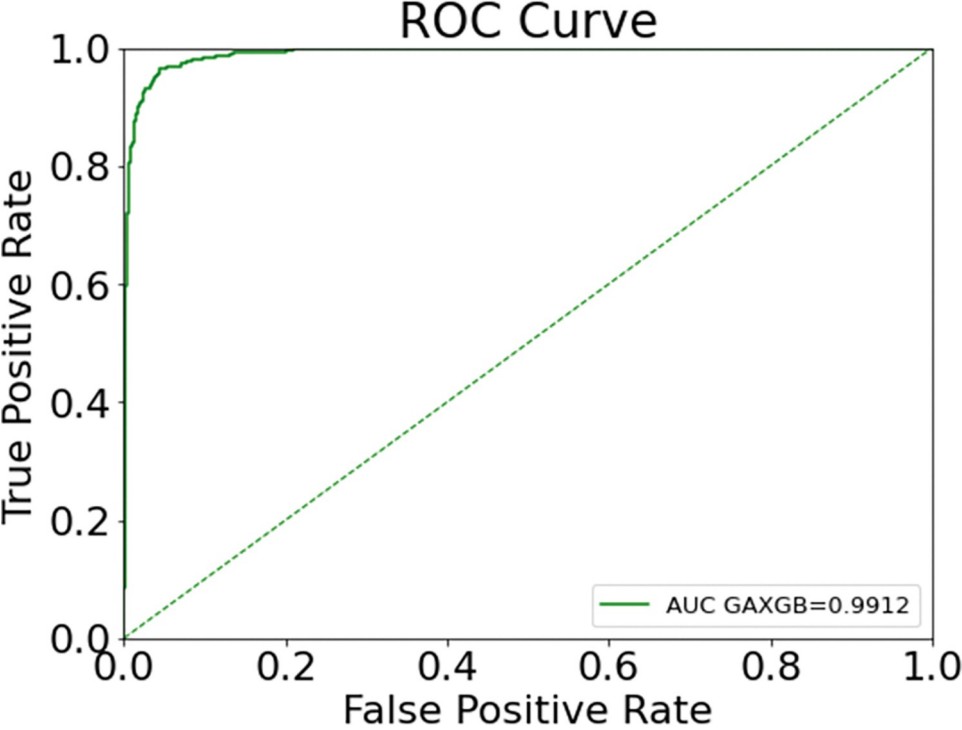

**Fig 9. ROC curve of GA-XGBoost.**

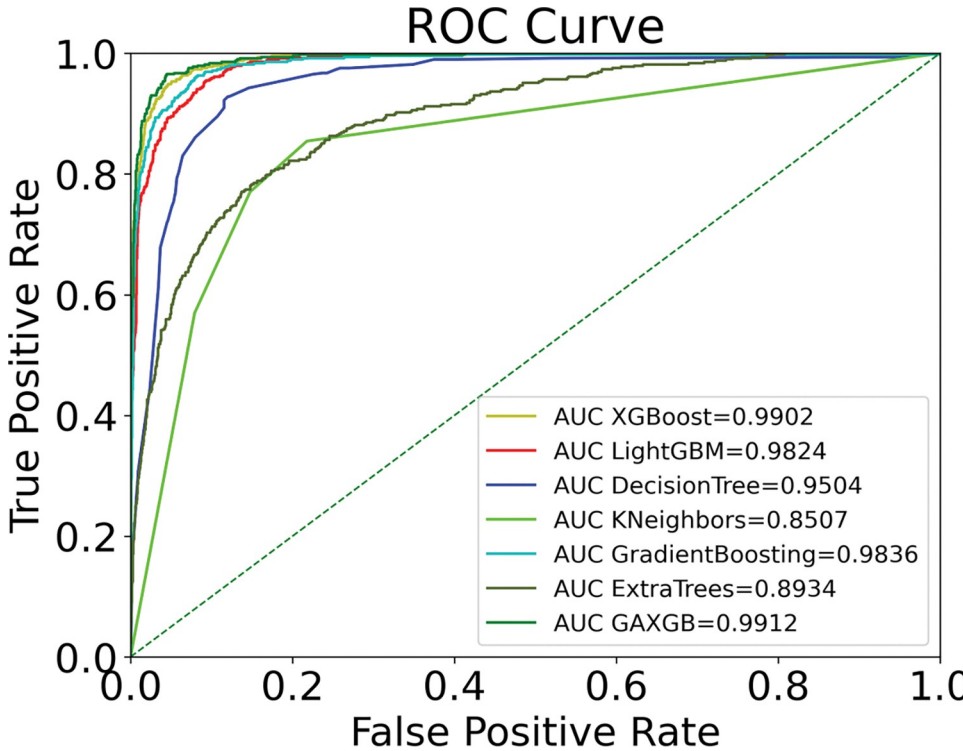

**Fig 10. ROC curve of models.**

**Table 9. Comparison of models test results.**

| Classifier | Precision | Recall | F1-score | Accuracy | AUC |
|---|---|---|---|---|---|
| XGBoost | 0.8488 | 0.9201 | 0.8830 | 0.9608 | 0.9902 |
| LightGBM | 0.7058 | 0.9242 | 0.8004 | 0.9260 | 0.9824 |
| DecisionTree | 0.6008 | 0.9098 | 0.7237 | 0.8885 | 0.9504 |
| KNN | 0.4993 | 0.7705 | 0.6060 | 0.8391 | 0.8507 |
| GBDT | 0.8167 | 0.9037 | 0.8580 | 0.9520 | 0.9836 |
| ExtraTrees | 0.4756 | 0.7971 | 0.5957 | 0.8263 | 0.8934 |
| **GA-XGBoost** | **0.8760** | **0.9262** | **0.9004** | **0.9671** | **0.9912** |

model output. A point represents a sample, and a red color indicates a larger feature value and a blue color indicates a smaller feature value. Fig 11 shows that Total_Trans_Ct is an important feature that is basically negatively correlated with the probability of churn. When the total number of transactions in the last year is low, the probability of churning is higher. On the other hand, when the total number of transactions in the last year is high, the probability of churn gradually decreases because the bank has multiple transactions with its customers and thus the level of trust increases. This characteristic of Total_Relationship_Count also has a significant impact on churn. The red dot area is mainly concentrated in the area where SHAP is less than zero, and the analysis shows that the more customers hold bank products, the less likely they are to churn. Total_Trans_Amt is positively correlated with SHAP, i.e. the higher the value, the higher the likelihood of churn. By analyzing the results of SHAP value, the total amount of transactions in the last year, due to the increase in the amount, customers may choose other products with higher returns and risks, resulting in lower sales of bank products.

Fig 12 shows the SHAP summary plot of the GA-XGBoost model feature importance ranking. This summary plot combines the importance of the features with the impact of the features. The position of the Y-axis is determined by the features that affect customer churn, and the position of the X-axis is determined by the Shapley value of each feature. From the figure, we can see that the three most important features in the model are Total_Trans_Ct, Total_Trans_Amt, and Total_Revolving_Bal. The Total_Trans_Ct feature has the greatest impact on the mode. The higher the value of this characteristic, the higher the Shapley value and the lower the probability of customer churn. The analysis of these key variables indicates that the number of transactions and the amount of transactions by users in the bank can be more effective in helping banks predict user churn. In addition, the results of the feature importance ranking found that the features of customer gender, presence of partners, and credit limit are low in influencing customer churn, and the bank can appropriately reduce the analysis of these features for influencing customer churn.

According to the feature importance ranking of SHAP in Fig 12 and the feature importance ranking of GA-XGBoost model in Fig 13, it can be seen that the ranking feature order is not exactly the same, and it can be concluded that the most critical factor influencing customer churn is Total_Trans_Ct, and both ranking algorithms rank the feature of Total_Trans_Ct as the factor influencing customer churn the most. In addition, both ranking algorithms rank Total_Trans_Ct, Total_Trans_Amt, Total_Revolving_Bal, and Total_Relationship_Count as the most important factors influencing customer churn. Through the comparative analysis of the characteristic importance ranking, the existing customers of the bank are less likely to churn the more bank products they hold and the more total transactions they make. Banks can launch specific financial products for this category of high-value users in the future customer relationship maintenance process, strengthen ties with them and involve this group in various banking operations and activities of the bank.

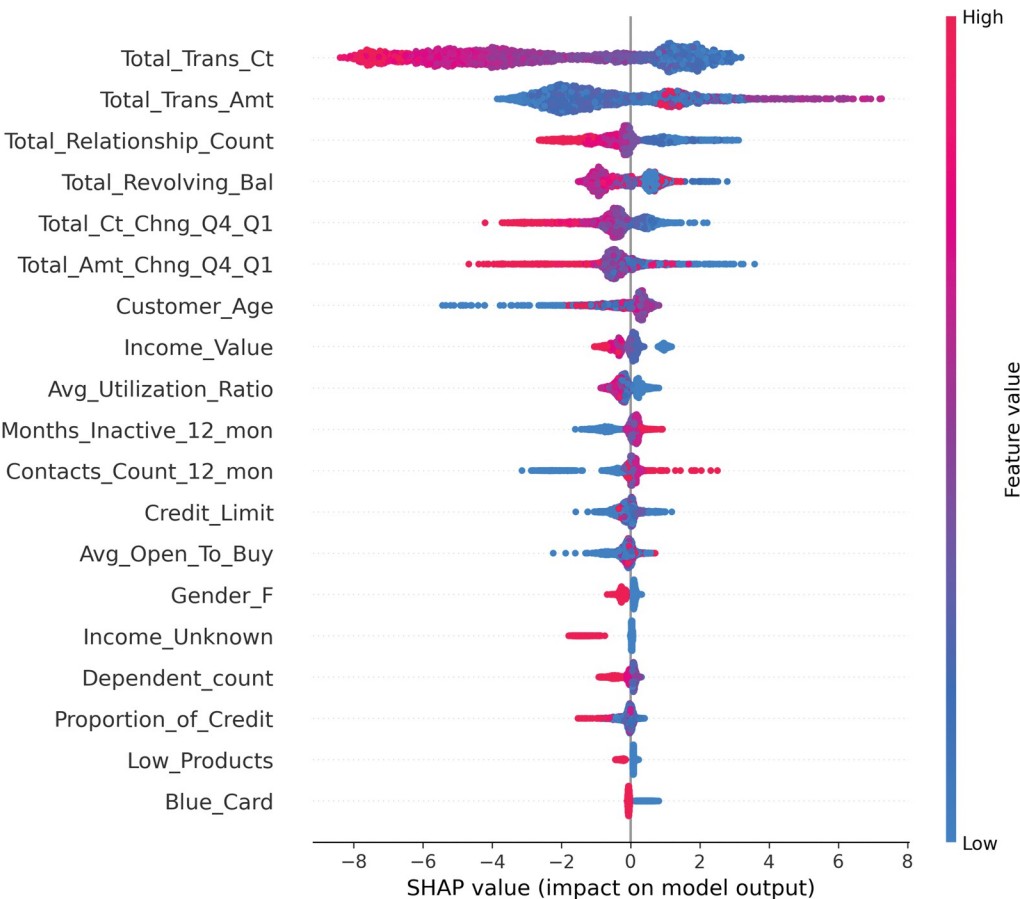

**Fig 11. Summary chart of SHAP feature analysis.**

**4.6.2 Partial explanation.**   For each sample that is predicted to be lost or not, the GA-XG-Boost model produces a predicted value, and the SHAP value is the value obtained from the sample features. As shown in Fig 14, for this sample of non-churning users, the values of Total_Trans_Ct and Total_Revolving_Bal are small and blue, indicating that these features reduce the SHAP value for this sample. A total of 87 transactions in the last year and a total working balance of $1,595 make it more likely that this user will continue to use the bank's products. However, it is also worth noting that the number of products held by the customer and the age of the customer also affect the likelihood of churn for this user. Since the number of products held by this user is 2 and his age is 47, the user can be considered a middle-aged customer who frequently uses the bank's transactions but is new to the bank's financial products according to the user profile.

As shown in Fig 15, for this sample of churned customers, the values of Total_Trans_Amt, Total_Trans_Ct, and Income-Category are large and red, indicating that these characteristics increase the SHAP value of this sample and play a positive role in the churn results. The bank can develop appropriate retention strategies for middle-aged customers with low income, such as offering products with high benefits and high cost performance, to attract them to participate in the bank's business, strengthen the relationship with them, and increase their loyalty to the bank. This group of customers can be attracted to participate in the bank's services, strengthen the relationship with customers, enhance their loyalty to the bank, and optimize marketing costs and resources.

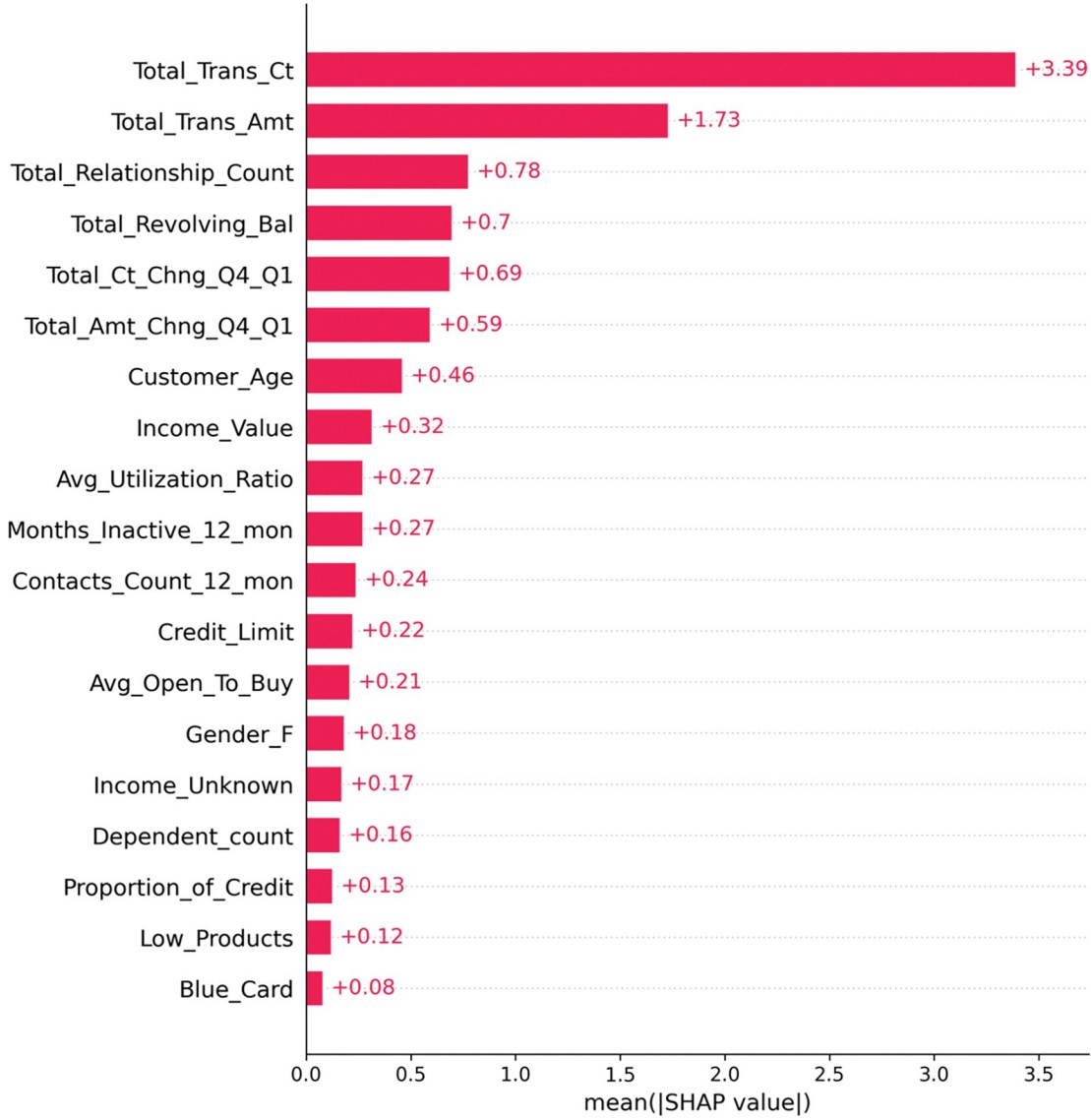

**Fig 12. SHAP summary plot of feature importance ranking of GA-XGBoost model.**

Since Total_Trans_Amt is strongly correlated with Total_Trans_Ct, Fig 16 shows the effect of Total_Trans_Ct and Total_Trans_Amt on predicting whether a customer will churn. The Shapley value is mostly positive until the Total_Trans_Ct is 60, and after the Total_Trans_Ct is 60, the Shapley value gradually decreases and becomes negative. The analysis of the SHAP values shows that the main effect of Total_Trans_Ct is negative, i.e. the probability of causing customer churn is lower when this characteristic is larger. The red dots appear more often when the SHAP value of Total_Trans_Ct is low, indicating a positive correlation between Total_-Trans_Ct and Total_Trans_Amt.

The SHAP waterfall plot of the churn prediction model for the 2nd user sample is shown in Fig 17, which illustrates how each feature drives the model output from the baseline value to the actual output value. In the figure, E[f(X)] is the average of all sample SHAPs with a value of 0.024; f(X) is the actual input SHAP value of the model in the current sample with a value of -8.547. By comparing f(X) < E[f(X)], it can be determined that the sample of users is an

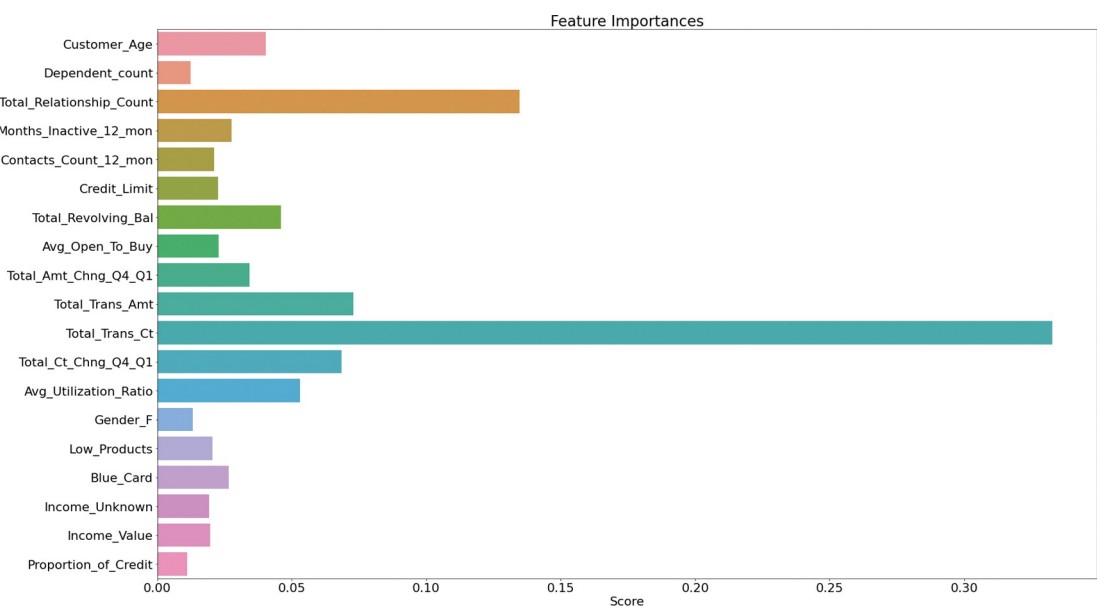

**Fig 13. GA-XGBoost feature importance order diagram.**

unchurned customer. The color of the SHAP value indicates the positive and negative impact of the sample on the predicted outcome, with blue indicating characteristics that decrease the predicted churn and red indicating characteristics that increase the predicted churn. In this user sample, Total_Trans_Ct is the most influential feature with a score of 87, causing a decrease in SHAP of 6.87. The second most important feature is Total_Revolving_Bal, which causes a decrease in SHAP of 0.98. In addition, the number of products owned by the customer is two and the age of the customer is 47, which increases the possibility of churning, but since the total SHAP value is negative, the model judges the user as a non-churning customer.

## 5 Discussion

In this paper, we train seven classifier models to predict the churn of potential bank customers. Our customer churn model construction steps include data preprocessing, data imbalance processing, machine learning model training and performance evaluation, hyperparameter tuning, and model interpretation. Through the experimental comparison analysis, GA-XG-Boost is the best bank customer churn prediction model, which effectively improves the prediction accuracy and precision by combining genetic algorithm and XGBoost algorithm. Moreover, the GA-XGBoost model uses the SHAP framework to explain how features in the data set specifically affect the customer churn results.

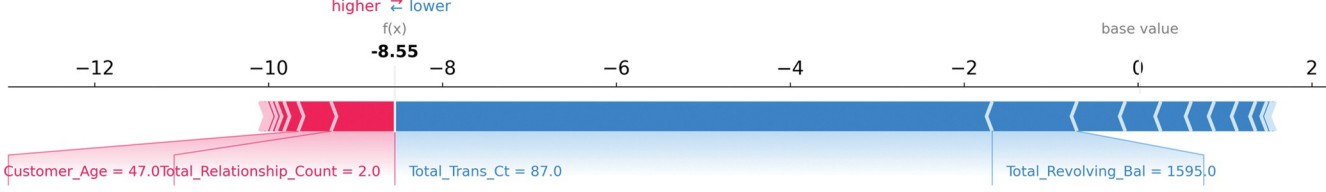

**Fig 14. Predicted effect of SHAP feature analysis for non-churning customers.**

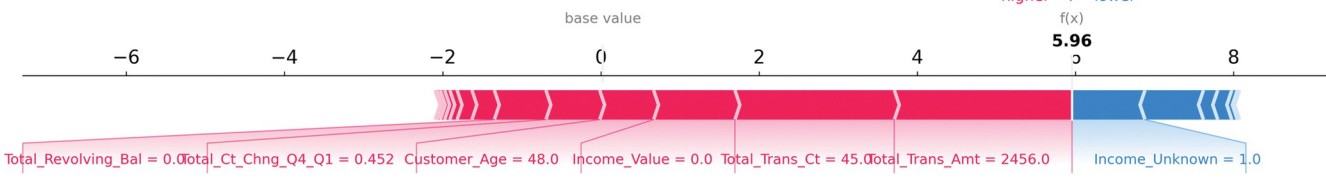

**Fig 15. Effect of SHAP feature analysis for predicted churn customers.**

Although some previous studies have also investigated the importance of the interpretation framework and parameter optimization for the model, for example, literature [13] and literature [14] have predicted telecom customer churn and used the corresponding interpretation model to interpret the classifier prediction results, they have only investigated a very limited interpretation approach without specifically explaining the features through both global and local interpretation levels, and they have not performed the parameter optimization for the specificity of the XGBoost algorithm. Although the literature [12–15] uses traditional grid search and random search for XGBoost parameter optimization, the experiments in this study show that genetic algorithms are more effective than traditional parameter optimization methods.

Table 5 shows the results of processing the unbalanced data. In the experiment, the data was balanced using SMOTE, ADASYN, and SMOTEENN. From the results of the experiments, it is clear that the model using the balanced data technique improves the performance of the model in a better way compared to the model that does not perform any manipulation on the data. The SMOTEENN resampling method gives the best performance in the XGBoost model when comparing several methods. The accuracy and recall of the model with the SMOTEENN method reach 96% and 92%, respectively.

Tables 7 and 8 show the results of parameter optimization. Hyperparameter tuning plays an important role in constructing models using ensemble learning approach, and genetic

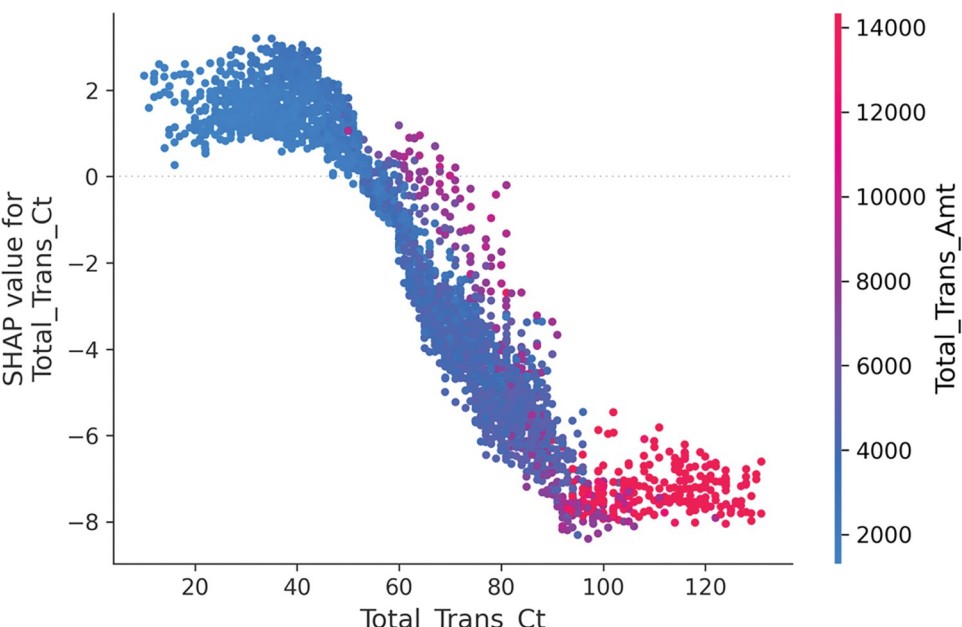

**Fig 16. SHAP feature dependence plot of Total_Trans_Ct and Total_Trans_Amt on model impact.**

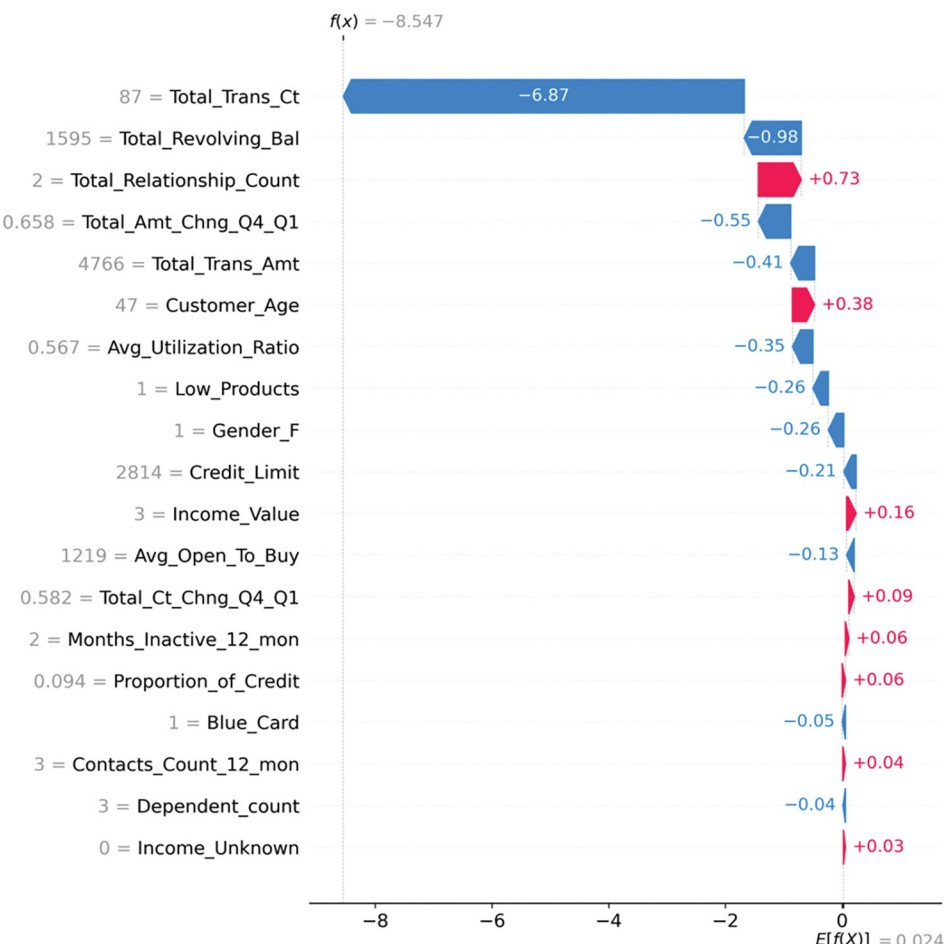

**Fig 17. Churn prediction SHAP waterfall for the 2nd user sample.**

algorithm is used as an improved solution in this study, and the outstanding improvement of genetic algorithm compared to traditional parameter optimization methods can be clearly reflected from the results. After data balancing of XGBoost using SMOTEENN, the parameters of XGBoost are optimized using genetic algorithm, and the final optimized parameters are n_estimators = 265, learning_rate = 0.097, max_depth = 5. From the recorded results, it is clear that the genetic algorithm improves XGBoost better than the grid search with the random search methods better improves the performance of the XGBoost model. From Fig 10, it can be seen that the GA-XGBoost model tuned by genetic algorithm achieves the highest AUC value, which reaches a value of 99.02%. The comparison of the model results in Table 9 shows that the GA-XGBoost model outperforms other machine learning models in every evaluation metric. It is clear that the application of the GA-XGBoost method significantly improves the performance of user churn prediction in the banking industry.

## 6 Conclusion

For the current research on customer churn prediction problem is mainly modeled by traditional machine learning methods, there are problems such as low prediction accuracy and poor generalization ability, this paper constructs an XGBoost model based on resampling and

genetic algorithm tuning method based on the original research, and selects a variety of traditional models for comparison. After that, in order to solve the black box problem of GA-XGBoost model, this paper further uses interpretable learning methods to explain the prediction results of the model, and the SHAP framework is used to explain and analyze the GA-XGBoost model as a whole and individually. Through the experiment, the following points were concluded from this research.

1. For the problem of unbalanced prediction categories in the data set, the model parameters were optimized by comparing multiple resampling methods and genetic algorithm, which is more in line with the unbalanced distribution of real bank data. In the difference significance test of the feature results, the p-value of significance is less than 0.05, indicating that there is indeed a fairly significant difference between the features and customer churn.

2. Compared with the traditional grid search method and random search algorithm for tuning, the genetic algorithm tuning is more effective in improving the prediction accuracy of the XGBoost model, and each evaluation index is improved to different degrees compared with the XGBoost model before tuning. The F1 and AUC values of the improved XGBoost model can reach 90% and 99%, respectively, which are higher than the other two machine learning models. It can be proved that the algorithm used in this study is optimal.

3. The SHAP visualization framework can solve the problem that integrated learning cannot predict the learning process, and it is beneficial for us to understand which features have a greater impact on the prediction results. In this study, the main factors affecting bank customer churn are the number of transactions in the last year, the amount of transactions in the last year, and the number of bank products held by the user through SHAP analysis; it explains that the prediction model, when predicting customer churn, has the same effect on the simple The portrait of churned customers is middle-aged customers aged 30–50 years old with higher and lower income levels. Banks respond to the problem of customer churn by focusing on the impact of number of transactions, deposit size and product type to identify customer groups with a tendency to churn in advance and effectively propose appropriate retention measures.

Although our model has a high accuracy and recall, there are still some feature redundancy and covariance problems that cause our model to be suboptimal. Our future research will be to use and compare different feature selection methods, such as combining RFE with GA-XGBoost, to select features that affect customer churn more and eliminate invalid features. In addition, since the dataset of this paper comes from the foreign Kaggle platform and lacks a wider range of application scenarios, we will collect real datasets from domestic banks for prediction in future practice to verify the effectiveness of the proposed method and develop corresponding retention strategies and marketing products for domestic bank employees to target the non-churning group.

## Author Contributions

**Conceptualization:** Ke Peng.

**Funding acquisition:** Yan Peng.

**Software:** Ke Peng.

**Validation:** Ke Peng, Yan Peng, Wenguang Li.

**Visualization:** Ke Peng.

**Writing – original draft:** Ke Peng, Yan Peng, Wenguang Li.

**Writing – review & editing:** Yan Peng.

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
