## [Decision Letter · Decision Letter 0]

13 Jun 2023

PONE-D-23-15889Research on Customer Churn Prediction and Model Interpretability AnalysisPLOS ONE

Dear Dr. Peng,

Thank you for submitting your manuscript to PLOS ONE. After careful consideration, we feel that it has merit but does not fully meet PLOS ONE’s publication criteria as it currently stands. Therefore, we invite you to submit a revised version of the manuscript that addresses the points raised during the review process.

We look forward to receiving your revised manuscript.

Kind regards,

Anas Bilal, Ph.D.

Academic Editor

PLOS ONE

Journal Requirements:

   "This work was supported in part by the Key Laboratory of Enterprise Informationization and Internet of Things Measurement and Control Technology in Sichuan Province Universities, No. 2021WYJ04; Sichuan University of Science and Engineering 2021 Postgraduate Innovation Fund Project, No. Y2021096."

Reviewers' comments:

Reviewer's Responses to Questions

**Comments to the Author**

1. Is the manuscript technically sound, and do the data support the conclusions?

Reviewer #1: Yes

Reviewer #2: Yes

2. Has the statistical analysis been performed appropriately and rigorously? 

Reviewer #1: No

Reviewer #2: Yes

3. Have the authors made all data underlying the findings in their manuscript fully available?

Reviewer #1: No

Reviewer #2: Yes

4. Is the manuscript presented in an intelligible fashion and written in standard English?

Reviewer #1: Yes

Reviewer #2: Yes

5. Review Comments to the Author

Reviewer #1: The paper will be accepted if the following comments are incorporated well.

1. Abstract needs to be more technical.

2. The contribution is unclear. Refer to the following paper, check how to specifically write the contribution at the end of Introduction section. “An Investigation of Credit Card Default Prediction in the Imbalanced Datasets”.

3. Clearly mention the research rap using latest research.

4. The flow of the paper is not good.

5. Consider the following paper to add in the separate literature review section: Spline-rule ensemble classifiers with structured sparsity regularization for interpretable customer churn modelling: An Analysis of Blessed Friday Sale at a Retail Store Using Classification Models: Customer Churn Prediction Model using Explainable Machine Learning: Corporate Bankruptcy Prediction: An Approach Towards Better Corporate World

6. Add one comparative analysis to validate that proposed technique significantly improved the performance of the models.

7. Authors should use some hypothesis testing to proved the developed models using different machine learning techniques are significantly same or different

8. Add the discussion section after results to prove the significance of the study.

9. Add the future directions and limitation of the study in the conclusion.

10. English grammar needs to be checked.

11. Overall speaking, the innovation points and main contributions of this paper need to be carefully reconsidered, and the innovation points should be presented more clear and prominent in terms of word expression and methodology & experiment design.

Reviewer #2: PONE-D-23-15889

Research on Customer Churn Prediction and Model Interpretability Analysis

how this paper aligns to the Plos One? what insights does it carry to the Plos One audiences. pls clearly mention them in the Introduction section.

Abstract and Keywords

pls cite a policy-implication sentence in the end of abstract.

1. INTRODUCTION

I would like to request to insert background and motivation, address gaps and laps, mention methodological and empirical contributions, policy implications, utility and applications of findings and contributions.

2. RELATED WORKS

add a new section, remove literature review from introduction then insert them in this section. clearly mentions what gaps and limitations you identified then what steps you taken to fill up those. 

3. Experimental Analysis

mention economic benefits of results and findings from three standpoint such as theoretical, empirical and marginal economic effect or, marginality analysis.

DISCUSSION

put a new section. add robustness test. compare your findings and contributions to those of relevant studies, mention your novelties.

how your findings and contributions could assist to mitigate market disaster comes from U-R conflicts and covid19.

5. Conclusion and Policy implications

summarize main results and findings, illustrate contributions.

comprehensively mention the policy implications, utility and application of this work, what are stakeholders' benefits to study this paper, mention with examples. what's the impact of this study?

it has typos and grammatical mistakes, pls do the professional proofreading.

References

delete low rated journals, cite 5 to 6 PLOS ONE papers.

6. PLOS authors have the option to publish the peer review history of their article (what does this mean?). If published, this will include your full peer review and any attached files.

Reviewer #1: No

Reviewer #2: No

---

## [Author Response · Author response to Decision Letter 0]

19 Jun 2023

Reviewer #1:

Comment 1:

Abstract needs to be more technical.

Author response 1: We appreciate your valuable comments and we have modified and adjusted the Abstract section accordingly. We supplement the abstract section with four aspects: background, methods, results and conclusions, respectively. The revised abstract is as follows: 

In recent years, with the continuous improvement of the financial system and the rapid development of the banking industry, the competition of the banking industry itself has intensified. At the same time, with the rapid development of information technology and Internet technology, customers' choice of financial products is becoming more and more diversified, and customers' dependence and loyalty to banking institutions is becoming less and less, and the problem of customer churn in commercial banks is becoming more and more prominent. How to predict customer behavior and retain existing customers has become a major challenge for banks to solve. Therefore, this study takes a bank's business data on Kaggle platform as the research object, uses multiple sampling methods to compare the data for balancing, constructs a bank customer churn prediction model for churn identification by GA-XGBoost, and conducts interpretability analysis on the GA-XGBoost model to provide decision support and suggestions for the banking industry to prevent customer churn. The results show that: (1) The applied SMOTEENN is more effective than SMOTE and ADASYN in dealing with the imbalance of banking data. (2) The F1 and AUC values of the model improved and optimized by XGBoost using genetic algorithm can reach 90% and 99%, respectively, which are optimal compared to other six machine learning models. The GA-XGBoost classifier was identified as the best solution for the customer churn problem. (3) Using Shapley values, we explain how each feature affects the model results, and analyze the features that have a high impact on the model prediction, such as the total number of transactions in the past year, the amount of transactions in the past year, the number of products owned by customers, and the total sales balance. The contribution of this paper is mainly in two aspects: (1) this study can provide useful information from the black box model based on the accurate identification of churned customers, which can provide reference for commercial banks to improve their service quality and retain customers; (2) it can provide reference for customer churn early warning models of other related industries, which can help the banking industry to maintain customer stability, maintain market position and reduce corporate losses. [Line 8-Line 27]

Comment 2:

The contribution is unclear. Refer to the following paper, check how to specifically write the contribution at the end of Introduction section. “An Investigation of Credit Card Default Prediction in the Imbalanced Datasets”.

Author response 2: Thanks for your excellent suggestion. We have added the contribution at the end of the introduction section. The specific contributions are as follows:

(1) First, by using multiple sampling techniques such as undersampling, oversampling and resampling to compare and contrast, the bank data imbalance problem is effectively solved.

(2) Second, based on the traditional XGBoost algorithm for customer churn, an improved XGBoost model is proposed, the main idea of which is to improve the overall performance of the model through hyper-parameter optimization. Genetic algorithm is used to optimize the parameters of the composite XGBoost algorithm after data equalization to obtain the bank customer churn prediction model, and then the sample categories are divided. The experimental results show that the improved XGBoost model significantly improves the predictive ability.

(3) Finally, the interpretable model is combined with the ensemble learning model, and the best prediction results of the ensemble learning model are interpreted globally and locally using the SHAP framework. This area is less analyzed by interpretable models, so this model can effectively help commercial bank decision makers to predict churning customers more accurately. In terms of its practical significance, when predicting bank customer churn, the SHAP value can visualize which characteristics mainly affect the customer churn outcome and whether the values of these characteristics increase the probability of customer churn, and then, based on the model prediction, facilitate bank staff to take relevant customer retention measures, which can further reduce bank customer churn, and also have certain reference significance for customer churn prediction in other fields. [Line 59-73]

Comment 3:

Clearly mention the research rap using latest research.

Author response 3: Thanks for your excellent suggestion. We have separated the previous literature review section from the introduction section and added the innovation points of this paper separately to the literature review section and compared it to recent years' customer churn research articles and methods. 

Comment 4:

The flow of the paper is not good.

Author response 4: We appreciate your valuable comments and we have replaced the previous chapter sections with the following chapter sections: 1. introduction 2. literature review 3. materials and methods 4. results 5. discussion 6. conclusion.

Comment 5:

Consider the following paper to add in the separate literature review section: Spline-rule ensemble classifiers with structured sparsity regularization for interpretable customer churn modelling: An Analysis of Blessed Friday Sale at a Retail Store Using Classification Models: Customer Churn Prediction Model using Explainable Machine Learning: Corporate Bankruptcy Prediction: An Approach Towards Better Corporate World.

Author response 5: We appreciate your valuable comments and we have added the corresponding literature in the literature review section and compared and contrasted the techniques used in these literatures with those used in this study, analyzed the deficiencies and limitations in these literatures, and then took measures to fill these deficiencies and limitations.

Comment 6:

Add one comparative analysis to validate that proposed technique significantly improved the performance of the models.

Author response 6: Thanks for your excellent suggestion. We have added the performance analysis and comparative analysis of the model in the results section to show the results of the model in figures and tables, and the results of the specific comparative analysis are as follows:

Finally, after the bank customer dataset was balanced by SMOTEENN, we built seven different machine learning models of LightGBM, DecisionTree, KNN, GBDT, ExtraTrees, XGBoost, and GA-XGBoost, respectively. The model comparison results are shown in Figure 10 and Table 9, where the precision, recall, F1 score, accuracy, and AUC of GA-XGBoost are 0.8760, 0.9262, 0.9004, 0.9671, and 0.9912, respectively, and each evaluation index is the highest among the seven different models. Through the experimental comparison, GA-XGBoost is the most effective in predicting bank customer churn, which can effectively help commercial banks to deal with customer churn in advance and make corresponding retention plans in real life. [Line 361-371]

Comment 7:

Authors should use some hypothesis testing to proved the developed models using different machine learning techniques are significantly same or different.

Author response 7: Thanks for your excellent suggestion. We have added a chi-square test for hypothesis testing in the data pre-processing section to calculate and analyze p-values for some of the features, setting the hypothesis whether these features are independent of customer churn or not, as shown in the following sections:

The chi-square test, which is a free distribution test, does not exist for specific parameters and the assumption of overall normal distribution, and the mean and variance cannot be calculated. For the bank customer dataset we used, we set H_0: the feature has no effect on customer churn results; H_1: the feature has an effect on customer churn results. Through the experimental results in Table 3, we can conclude that the p-values of Gender, Income_Category, and Education_Level are almost less than 0.05, and the original hypothesis H_0 is rejected, that is These features have a significant effect on customer churn results. p-value indicates the probability of mutual independence when the sample row variables and column variables are drawn from the overall, and since the probability value of p is small, we reject the original hypothesis that the features and customer churn results are independent of each other.

[Line 249-258]

Comment 8:

Add the discussion section after results to prove the significance of the study.

Author response 8: Thanks for your excellent suggestion. We have added a discussion section after the results section to demonstrate the significance of the study and to compare my findings and contributions with related studies, mentioning the importance of my findings and contributions. [Line 455-477]

Comment 9:

Add the future directions and limitation of the study in the conclusion.

Author response 9: We appreciate your valuable comments and we add the future directions and limitation of the study in the conclusion. The specific future directions and limitation are as follows:

Although our model has a high accuracy and recall, there are still some feature redundancy and covariance problems that cause our model to be suboptimal. Our future research will be to use and compare different feature selection methods, such as combining RFE with GA-XGBoost, to select features that affect customer churn more and eliminate invalid features. In addition, since the dataset of this paper comes from the foreign Kaggle platform and lacks a wider range of application scenarios, we will collect real datasets from domestic banks for prediction in future practice to verify the effectiveness of the proposed method and develop corresponding retention strategies and marketing products for domestic bank employees to target the non-churning group. [Line 503-509]

Comment 10:

English grammar needs to be checked.

Author response 10: Thanks for your excellent suggestion. We have used the appropriate software to test and revise the English grammar and hope to meet the requirements.

Comment 11:

Overall speaking, the innovation points and main contributions of this paper need to be carefully reconsidered, and the innovation points should be presented more clear and prominent in terms of word expression and methodology & experiment design.

Author response 11: Thanks for your excellent suggestion. We have mentioned the innovative points of this paper several times in the experimental section and in the methodological aspect, and we have added a contribution section at the end of the introduction section to specify the innovative points as well as the significance of this paper.

Reviewer #2:

Comment 1:

how this paper aligns to the Plos One? what insights does it carry to the Plos One audiences. pls clearly mention them in the Introduction section.

Author response 1: Thanks for your excellent suggestion. We have added the methodology and significance of the paper in the introduction section and have written the contributions specifically at the end of the introduction section. The specific contents are as follows:

This research aims to analyze and study the current status of customer churn in commercial banks and identify the causes of the current churn. The GA-XGBoost algorithm is used to build a bank customer churn prediction model and to explain and analyze the causes of customer churn as well as customer retention strategies by relying on interpretability related theories. Based on the study of data mining related technology theories and theories related to GA-XGBoost algorithm in customer churn application, the raw data of a bank is analyzed.

The main contributions of this paper are as follows:

(1) First, by using multiple sampling techniques such as undersampling, oversampling and resampling to compare and contrast, the bank data imbalance problem is effectively solved.

(2) Second, based on the traditional XGBoost algorithm for customer churn, an improved XGBoost model is proposed, the main idea of which is to improve the overall performance of the model through hyper-parameter optimization. Genetic algorithm is used to optimize the parameters of the composite XGBoost algorithm after data equalization to obtain the bank customer churn prediction model, and then the sample categories are divided. The experimental results show that the improved XGBoost model significantly improves the predictive ability.

(3) Finally, the interpretable model is combined with the ensemble learning model, and the best prediction results of the ensemble learning model are interpreted globally and locally using the SHAP framework. This area is less analyzed by interpretable models, so this model can effectively help commercial bank decision makers to predict churning customers more accurately. In terms of its practical significance, when predicting bank customer churn, the SHAP value can visualize which characteristics mainly affect the customer churn outcome and whether the values of these characteristics increase the probability of customer churn, and then, based on the model prediction, facilitate bank staff to take relevant customer retention measures, which can further reduce bank customer churn, and also have certain reference significance for customer churn prediction in other fields. [Line 58-78]

Comment 2:

I would like to request to insert background and motivation, address gaps and laps, mention methodological and empirical contributions, policy implications, utility and applications of findings and contributions.

Author response 2: Thanks for your excellent suggestion. We have revised the introduction section to include four specific sections: the background of customer churn, the significance and impact of the methods used in the paper, the main contributions of the paper, and the structure of the paper. [Line 30-83]

Comment 3:

add a new section, remove literature review from introduction then insert them in this section. clearly mentions what gaps and limitations you identified then what steps you taken to fill up those. 

Author response 3: Thanks for your excellent suggestion. We have separated the previous literature review section from the introduction section and added the innovation points of this paper separately to the literature review section and compared it to recent years' customer churn research articles and methods. [Line 86-133]

Comment 4:

mention economic benefits of results and findings from three standpoint such as theoretical, empirical and marginal economic effect or, marginality analysis. 

Author response 4: Thanks for your excellent suggestion. We have divided the experimental analysis into a methodological part (including theoretical elaboration, data preprocessing and hypothesis testing, and experimental evaluation criteria) and a results part (including data equilibration processing, model construction, and parameter optimization), and compared the performance analysis of the models used.

Comment 5:

put a new section. add robustness test. compare your findings and contributions to those of relevant studies, mention your novelties.

how your findings and contributions could assist to mitigate market disaster comes from U-R conflicts and covid19.

Author response 5: Thanks for your excellent suggestion. We have added a discussion section in the next chapter of the results section, specifically comparing my findings and contributions to related research, mentioning what you have innovated. As well as describing how my findings and contributions can help banks retain customers that are about to be lost. [Line 460-482]

Comment 6:

summarize main results and findings, illustrate contributions.

comprehensively mention the policy implications, utility and application of this work, what are stakeholders' benefits to study this paper, mention with examples. what's the impact of this study?

Author response 6: Thanks for your excellent suggestion. I have stated the research implications of this study at the end of the discussion section and added directions and limitations for future research in the conclusion section. The specific research implications are as follows:

In this paper, a bank customer churn prediction model was constructed using GA-XGBoost, and the SHAP framework was used to explain how characteristics affect customer churn outcomes. By constructing a churn early warning model and predicting potential churn customer groups, it provides some data and theoretical support for commercial banks' subsequent customer relationship management, which can help the banking industry maintain customer stability, maintain corporate market position and reduce corporate losses, as well as promote the benign development of the domestic banking industry and the continuous improvement and innovation of data mining technology. At the same time, these theoretical discussions also provide references for customer churn early warning models in other related industries. [Line 460-514]

Comment 7:

it has typos and grammatical mistakes, pls do the professional proofreading..

Author response 7: Thanks for your excellent suggestion. We have used the appropriate software to test and revise the English grammar and hope to meet the requirements.

---

## [Decision Letter · Decision Letter 1]

19 Jul 2023

PONE-D-23-15889R1Research on Customer Churn Prediction and Model Interpretability AnalysisPLOS ONE

Dear Dr. Peng,

Thank you for submitting your manuscript to PLOS ONE. After careful consideration, we feel that it has merit but does not fully meet PLOS ONE’s publication criteria as it currently stands. Therefore, we invite you to submit a revised version of the manuscript that addresses the points raised during the review process.

We look forward to receiving your revised manuscript.

Kind regards,

Anas Bilal, Ph.D.

Academic Editor

PLOS ONE

Reviewers' comments:

Reviewer's Responses to Questions

**Comments to the Author**

1. If the authors have adequately addressed your comments raised in a previous round of review and you feel that this manuscript is now acceptable for publication, you may indicate that here to bypass the “Comments to the Author” section, enter your conflict of interest statement in the “Confidential to Editor” section, and submit your "Accept" recommendation.

Reviewer #1: All comments have been addressed

Reviewer #3: All comments have been addressed

2. Is the manuscript technically sound, and do the data support the conclusions?

Reviewer #1: Yes

Reviewer #3: Yes

3. Has the statistical analysis been performed appropriately and rigorously? 

Reviewer #1: Yes

Reviewer #3: Yes

4. Have the authors made all data underlying the findings in their manuscript fully available?

Reviewer #1: Yes

Reviewer #3: Yes

5. Is the manuscript presented in an intelligible fashion and written in standard English?

Reviewer #1: Yes

Reviewer #3: Yes

6. Review Comments to the Author

Reviewer #1: The manuscript is acceptable in the current form as the comments are responded in an effective way.

Reviewer #3: author has worked on various sections to improve the paper. There are improvement from the previous attempt. However, i would still suggest to improve the discussion section.

7. PLOS authors have the option to publish the peer review history of their article (what does this mean?). If published, this will include your full peer review and any attached files.

Reviewer #1: No

Reviewer #3: **Yes: **Dr. Muhammad Usman

---

## [Author Response · Author response to Decision Letter 1]

20 Jul 2023

Comment 1:

author has worked on various sections to improve the paper. There are improvement from the previous attempt. However, i would still suggest to improve the discussion section.

Author response 1: We appreciate your valuable comments and we have modified and adjusted the Discussion section accordingly. We have added a discussion section describing the results of the experiment as follows. 

Table 5 shows the results of processing the unbalanced data. In the experiment, the data was balanced using SMOTE, ADASYN, and SMOTEENN. From the results of the experiments, it is clear that the model using the balanced data technique improves the performance of the model in a better way compared to the model that does not perform any manipulation on the data. The SMOTEENN resampling method gives the best performance in the XGBoost model when comparing several methods. The accuracy and recall of the model with the SMOTEENN method reach 96% and 92%, respectively.

Tables 7 and 8 show the results of parameter optimization. Hyperparameter tuning plays an important role in constructing models using ensemble learning approach, and genetic algorithm is used as an improved solution in this study, and the outstanding improvement of genetic algorithm compared to traditional parameter optimization methods can be clearly reflected from the results. After data balancing of XGBoost using SMOTEENN, the parameters of XGBoost are optimized using genetic algorithm, and the final optimized parameters are n_estimators=265, learning_rate=0.097, max_depth=5. From the recorded results, it is clear that the genetic algorithm improves XGBoost better than the grid search with the random search methods better improves the performance of the XGBoost model. From Figure 10, it can be seen that the GA-XGBoost model tuned by genetic algorithm achieves the highest AUC value, which reaches a value of 99.02%. The comparison of the model results in Table 9 shows that the GA-XGBoost model outperforms other machine learning models in every evaluation metric. It is clear that the application of the GA-XGBoost method significantly improves the performance of user churn prediction in the banking industry.

---

## [Editor Report · Decision Letter 2]

25 Jul 2023

Research on Customer Churn Prediction and Model Interpretability Analysis

PONE-D-23-15889R2

Dear Dr. Ke,

We’re pleased to inform you that your manuscript has been judged scientifically suitable for publication and will be formally accepted for publication once it meets all outstanding technical requirements.

Kind regards,

Anas Bilal, Ph.D.

Academic Editor

PLOS ONE

---

## [Editor Report · Acceptance letter]

27 Jul 2023

PONE-D-23-15889R2 

Research on Customer Churn Prediction and Model Interpretability Analysis 

Dear Dr. Peng:

I'm pleased to inform you that your manuscript has been deemed suitable for publication in PLOS ONE. Congratulations! Your manuscript is now with our production department. 

Kind regards, 

on behalf of

Dr. Anas Bilal 

Academic Editor

PLOS ONE